# TORC1 Signaling Controls the Stability and Function of α-Arrestins Aly1 and Aly2

**DOI:** 10.3390/biom12040533

**Published:** 2022-03-31

**Authors:** Ray W. Bowman, Eric M. Jordahl, Sydnie Davis, Stefanie Hedayati, Hannah Barsouk, Nejla Ozbaki-Yagan, Annette Chiang, Yang Li, Allyson F. O’Donnell

**Affiliations:** 1Department of Biological Sciences, University of Pittsburgh, Pittsburgh, PA 15260, USA; rwb42@pitt.edu (R.W.B.II); emj46@pitt.edu (E.M.J.); skd37@pitt.edu (S.D.); sth69@pitt.edu (S.H.); hannah.barsouk@yale.edu (H.B.); neo10@pitt.edu (N.O.-Y.); anc64@pitt.edu (A.C.); 2Department of Cell Biology, School of Medicine, University of Pittsburgh, Pittsburgh, PA 15261, USA; yangli@pitt.edu

**Keywords:** kinase, phosphatase, *Saccharomyces cerevisiae*, protein trafficking, ubiquitin, fluorescence microscopy, α-arrestin, vacuole proteases, protein degradation

## Abstract

Nutrient supply dictates cell signaling changes, which in turn regulate membrane protein trafficking. To better exploit nutrients, cells relocalize membrane transporters via selective protein trafficking. Key in this reshuffling are the α-arrestins, selective protein trafficking adaptors conserved from yeast to man. α-Arrestins bind membrane proteins, controlling the ubiquitination and endocytosis of many transporters. To prevent the spurious removal of membrane proteins, α-arrestin-mediated endocytosis is kept in check through phospho-inhibition. This phospho-regulation is complex, with up to 87 phospho-sites on a single α-arrestin and many kinases/phosphatases targeting α-arrestins. To better define the signaling pathways controlling paralogous α-arrestins, Aly1 and Aly2, we screened the kinase and phosphatase deletion (KinDel) library, which is an array of all non-essential kinase and phosphatase yeast deletion strains, for modifiers of Aly-mediated phenotypes. We identified many Aly regulators, but focused our studies on the TORC1 kinase, a master regulator of nutrient signaling across eukaryotes. We found that TORC1 and its signaling effectors, the Sit4 protein phosphatase and Npr1 kinase, regulate the phosphorylation and stability of Alys. When Sit4 is lost, Alys are hyperphosphorylated and destabilized in an Npr1-dependent manner. These findings add new dimensions to our understanding of TORC1 regulation of α-arrestins and have important ramifications for cellular metabolism.

## 1. Introduction

Cells respond to environmental changes, including nutrient fluctuations, by regulating the complement of membrane proteins at the cell surface. How is this selective reshuffling of the plasma membrane (PM)-embedded proteome achieved in response to stress or nutrient cues? In *Saccharomyces cerevisiae*, a powerful model system for protein trafficking studies [1,2], selective protein trafficking adaptors known as the α-arrestins (i.e., arrestin-related trafficking adaptors (ARTs)) are molecular kingpins in this process [3,4,5,6,7]. Under rigorous regulation by phospho-signaling networks, including the two major nutrient and energy-sensing kinase complexes—the target of rapamycin complex 1, or TORC1, and the AMP-activated protein kinase (AMPK), known as Snf1 in yeast—, the α-arrestins respond to environmental fluctuations by selectively binding to an ever-expanding repertoire of membrane proteins, including glucose, amino acid, inositol, and metal ion transporters, as well as G-protein coupled receptors (GPCRs) [4,5,7,8,9,10,11,12,13]. To remove these membrane proteins from the PM, the α-arrestins rely on their association with the E3 ubiquitin (Ub) ligase, Rsp5. The α-arrestins act as a bridge between the nutrient transporters and Rsp5, bringing them near one another. Ubiquitination of transporters by Rsp5 creates a new binding interface for their association with components of the endocytic machinery, ensuring membrane protein clustering and efficient endocytosis [14,15,16,17]. This paradigm for endocytic regulation by α-arrestins is closely mirrored by their well-studied relatives, the β-arrestins, which, in mammals, control the endocytosis and intracellular sorting of many membrane proteins, including GPCRs [18,19,20]. Though best characterized in yeast, the orthologous mammalian α-arrestins (i.e., arrestin-domain-containing proteins (ARRDCs)) share conserved roles in controlling nutrient transporter and GPCR trafficking and activity [21,22,23,24,25].

To avoid the spurious removal of membrane proteins at the PM, α-arrestin-mediated endocytosis is carefully regulated in cells. Three mechanisms of α-arrestin regulation have emerged: (1) the activity of α-arrestins is controlled by phosphorylation, which typically impedes the endocytic activity of these adaptors but may stimulate intracellular sorting activities [5,8,12,26,27,28,29,30,31]; (2) the stability of α-arrestins is controlled and α-arrestins are degraded when cells are exposed to conditions in which they are no longer needed [29,32,33]; (3) the expression of some α-arrestin-encoding genes is condition-specific (i.e., the genes for α-arrestins Ecm21/Art2 or Csr2/Art8 are expressed in nitrogen or glucose starvation conditions, respectively) [9,34]. Here, we focused on identifying mechanisms of phospho-regulation for α-arrestins Aly1 and Aly2 (arrestin-like yeast proteins 1 and 2) and found it to be tightly associated with Aly protein stability. This is an established paradigm for α-arrestins, where phosphorylation of α-arrestin Rog3 by the Snf1 kinase causes Rog3 destabilization [29]. However, the pathway needed for Snf1-regulated destruction of Rog3 remains to be fully defined. Key to the phospho-inhibition of α-arrestins is their sequestration by 14-3-3 proteins. When phosphorylated, α-arrestins Bul1, Bul2, and Rod1 bind to 14-3-3 proteins, and are unable to regulate the endocytosis of the general amino acid permease, Gap1, or the monocarboxylate transporter, Jen1 [12,31]. Directed or large-scale proteomics approaches find that most α-arrestins bind 14-3-3 proteins, supporting this as a possibly widespread regulatory mechanism [12,28,31,34,35,36].

With between 13 and 87 α-arrestin phosphorylation sites identified to date, either by targeted or global proteomics approaches [8,28,37,38,39,40,41], the network of kinases and phosphatases responsible for modifying these trafficking adaptors is likely extensive. Given the complexity of phospho-regulation and its ramifications for α-arrestin function and downstream cellular metabolism, we used a targeted genetic screen to identify phospho-regulators of α-arrestins Aly1 and Aly2. We examined the impact of each non-essential kinase or phosphatase in regulating Aly-mediated growth on: (1) rapamycin, a drug that inhibits the nutrient-sensing TORC1 kinase complex, or (2) high salt stress, which triggers activation of several kinase and phosphatases in yeast, including the high osmolarity glycerol response kinase, Hog1, and the calcineurin (CN) protein phosphatase [42,43,44,45]. We chose these two conditions because TORC1 and CN control Aly function. In addition, high salt conditions trigger the internalization of several Aly-regulated membrane proteins, including the glutamic and aspartic acid permease, Dip5, and a P-type ATPase sodium pump, Ena1 [46]. We identify an expansive phospho-regulatory network of over 60 kinases/phosphatases altering these Aly-dependent phenotypes. We focused on 18 kinases and phosphatases from this initial screen and were surprised to find that nearly 90% of these were important for maintaining Aly1 and/or Aly2 protein levels, suggesting a possible wide-spread connection between phospho-regulation and Aly stability.

Our screen identified components of the yeast mating pathway. Alys or other α-arrestins control the trafficking of Ste3 and Ste2, respectively, which are the GPCRs that regulate mating response in *MAT***a** or *MAT*α haploid cells [10,11]. However, we chose to pursue the genetic ties linking Alys to the TORC1-Sit4-Npr1 signaling cascade. Specifically, Sit4, which is a ceramide-activated, type2A-related protein phosphatase in yeast, was a strong candidate in our genetic screen, impairing the ability of Aly1 and Aly2 to improve growth on rapamycin. There are many Sit4-α-arrestin ties already forged in the literature [12,13,32]. In the TORC1-Sit4-Npr1 pathway, TORC1 inhibition (as occurs in rapamycin-treated or nitrogen-starved cells) releases Sit4 from its TORC1- and Tap42-bound inhibitory complex, where Tap42 is an essential protein, binding partner to Sit4, and regulator of TORC1 signaling [47,48]. Also identified in our study, Tip41 prevents the inhibitory Tap42-Sit4 complex from forming and promotes Sit4 association with Sit4-associated proteins (SAPs), which help direct Sit4 to specific cellular targets [12,13,47]. Downstream of Sit4, the nitrogen permease reactivator kinase, Npr1, regulates α-arrestin phosphorylation [5,8,49]. Once dissociated from TORC1, Sit4 dephosphorylates Npr1, converting it to an active kinase [47,50]. When TORC1 activity is high, as it is in cells grown in the presence of abundant amino acids and nutrients, Npr1 is hyperphosphorylated, and is thought to be largely inactive. Npr1 activity is therefore controlled by a balance of TORC1-directed phosphorylation and Sit4-mediated dephosphorylation, among other regulatory inputs [8,51]. In *sit4*∆ cells, Npr1 is hyperphosphorylated and thought to be largely inactive [47,49]. However, recent studies have expanded on this TORC1-Sit4-Npr1 model, suggesting a more nuanced relationship between these signaling factors [52,53]. Relevant here, Npr1 phosphorylates some targets even when TORC1 is active [37]. Sit4 control is also more complex, with Sit4 regulating some substrates even when TORC1 is active [12,13,52,53]. Indeed, there is evidence that, in conditions where Sit4 would be bound to TORC1, it selectively targets the α-arrestins for dephosphorylation [12,13].

Thus, Npr1 and Sit4 both regulate α-arrestins. Importantly for our studies, Npr1 directly phosphorylates Aly2 [5]. Aly1 phosphorylation changes during Npr1 inhibition, which is consistent with it also being an Npr1 substrate [37]. Npr1 is required for Aly2 to induce the trafficking of Gap1 to the PM under nutrient-limiting conditions [5]. In contrast, Npr1 phosphorylation of α-arrestins Bul1 and Bul2 inhibits their ability to stimulate the endocytosis of the general amino acid permease, Gap1 [12,13]. Dephosphorylation of Bul1 and Bul2 is controlled by Sit4, with active Sit4 dephosphorylating Buls under starvation conditions, allowing them to stimulate endocytic trafficking of Gap1 and Jen1 [12,13,27,34]. A similar model has been proposed for α-arrestin Ldb19/Art1, with Npr1 phosphorylating Art1 to block Art1-mediated endocytosis of the arginine permease, Can1, in response to rapamycin [8]. However, in another study, cells lacking Npr1 retained Can1 at the PM in response to rapamycin, which raises the possibility that this regulation is more nuanced and merits further exploration [54]. Finally, α-arrestin Art2 induces internalization of Thi7, a thiamine transporter, in response to thiamine and cycloheximide (CHX) treatment, and, in this process, Art2 is positively regulated by Sit4, though is largely independent of Npr1 [32].

We find that Aly1 and Aly2 are hyperphosphorylated in cells lacking the Sit4 phosphatase. Aly hyperphosphorylation occurs in rich growth medium, suggesting that, even when TORC1 is active, either the Sit4 phosphatase is dephosphorylating Alys or a kinase under the control of Sit4 is hyperactivated and leads to α-arrestin hyperphosphorylation. Further, Aly protein abundance is diminished in *sit4*∆ cells, thus linking hyperphosphorylation with the instability/degradation of Alys. This is consistent with the phospho-induced instability of other α-arrestins [29]. Herein, we worked to define the mechanism of α-arrestin degradation in cells lacking *sit4*, and found that the vacuole regulates Aly degradation. Interestingly, further deletion of the Npr1 kinase in *sit4*∆ cells reversed the hyperphosphorylation of Alys, improved their stability, and restored trafficking of the glycerophosphoinositol transporter, Git1, to the vacuole. Git1 trafficking is regulated by Alys [55], and so the protein-trafficking function of α-arrestins is under the control of the TORC1-Sit4-Npr1 pathway. We propose a model for TORC1-Sit4-Npr1 regulation of Alys, wherein Npr1 is needed for the hyperphosphorylation of Alys, which stimulates their degradation. Npr1 regulation is antagonized by the Sit4 phosphatase, which either directly dephosphorylates Alys or indirectly counters Aly phosphorylation by influencing Npr1. In contrast to other α-arrestin studies [8,12,13], under nutrient-replete conditions where TORC1 is active and Npr1 is generally thought of as inactive, the Alys were still hyperphosphorylated in an Npr1-dependent manner when the Sit4 phosphatase was lost. We suggest that Npr1 activity, under these conditions, while dampened with respect to some substrates, is maintained towards the Alys. However, the Npr1 kinase is kept in check by the Sit4 phosphatase, preserving a pool of dephosphorylated and active α-arrestins. This balance of kinase and phosphatase activity creates a phospho-cycle that serves as a rheostat for α-arrestin-mediated endocytosis, even under nutrient-replete conditions.

## 2. Materials and Methods

### 2.1. Yeast Strains and Growth Conditions

The yeast strains used in this study are described in Appendix A and are derived from the BY4741 or BY4742 genetic backgrounds of *S. cerevisiae* (S288C in origin). Yeast cells were grown in either synthetic complete (SC) medium lacking the appropriate amino acid for plasmid selection, prepared as described in [56] and using ammonium sulfate as a nitrogen source, or YPD medium where indicated. Unless otherwise indicated, yeast cells were grown at 30 °C. Liquid medium was filter-sterilized and solid medium for agar plates had 2% agar *w*/*v* added before autoclaving.

### 2.2. Serial Dilution Growth Assays

For the serial dilution growth assays on solid medium, cells were grown to saturation in liquid SC medium overnight, and the A_600_ was determined. Starting with an A_600_ of 1.0 (approximately 1.0 × 10^7^ cells/mL), serial dilutions of either 3-fold (as in Appendix A) or 5-fold (as in Figure 1A,B) were then generated and transferred to solid medium using a sterile replica-pinning tool. Cells were then grown at 30 °C for 3–6 days and images were captured using a Chemidoc XRS+ imager (BioRad, Hercules, CA, USA) and all were evenly modified in Photoshop (Adobe Systems Incorporated, San Jose, CA, USA). For rapamycin-containing plates, the rapamycin was obtained from LC Laboratories (Woburn, MA, USA), and a stock solution of 0.5 mg/mL in ethanol was made and then diluted to the final concentration indicated in the medium in each figure panel (typically 50 ng/mL). For salt-containing medium, sodium chloride was dissolved in the SC medium prior to autoclaving to generate the final concentrations indicated in each figure.

### 2.3. Plasmids and DNA Manipulations

Plasmids used in this work are described in Appendix A. PCR amplifications for generating plasmid constructs were performed using Phusion High Fidelity DNA polymerase (ThermoFisher Scientific, Waltham, MA, USA) and confirmed by DNA sequencing. Plasmids were transformed into yeast cells using the lithium acetate method [57] and transformants were selected for use on SC mediums lacking specific nutrients.

### 2.4. KinDel Library Screen

The kinase and phosphatase deletion (KinDel) library contains 187 unique non-essential gene deletions, each of which is annotated as being non-essential in the *Saccharomyces*
Genome Database (SGD). This library initiated with a collection generated by Dr. J. Patterson (Patterson and Thorner, unpublished). We then used YeastMine, populated by SGD and powered by InterMine, and searched for ‘protein kinase’ or ‘protein phosphatase’. These searches returned nearly 3000 candidates before duplicates and essential genes were removed. SGD annotations of the remaining genes were then further assessed for protein kinase and phosphatase function, resulting in a final list of 187 gene deletion mutants, which were then arrayed over two 96-well plates. The library is available upon request and is further documented at https://www.odonnelllab.com/yeast-libraries (accessed on 23 February 2022).

To screen for modifiers of α-arrestins’ Aly1 and Aly2 function, the KinDel library was transformed with either pRS426-vector or pRS426-Aly1 or -Aly2 plasmids, each of which expressed Aly1 or Aly2 from their endogenous promoter but on a 2-micron plasmid backbone. Transformations were performed as described in [57] and were done with the aid of the Benchtop RoToR HAD robotic plate handler (Singer Instruments Co. Ltd., Roadwater, UK) and the Multidrop Combi (ThermoFisher) in the laboratory of Dr. Anne-Ruxandra Carvunis (Univ. of Pittsburgh). Each transformed version of the library was stamped in technical triplicate to SC medium lacking uracil (as a control), SC-Ura- medium containing 0.8M NaCl, 1.5M NaCl, or 50 ng/mL rapamycin. Plates were grown at 30 °C for 2–4 days, and white light images were captured using the BioRad ChemiDoc XRS+ imager (Hercules, CA, USA) on days 1–4 of incubation. Images were converted to .jpg file format, and the pixel size of colonies was measured using the DissectionReader macro (generously provided by Dr. Kara Bernstein’s laboratory (Univ. of Pittsburgh) and developed by John C. Dittmar and Robert J.D. Reid in Dr. Rodney Rothstein’s laboratory (Columbia Univ.)) in Image J (National Institutes of Health, Bethesda, MA, USA). More information on this plugin can be found at: https://github.com/RothsteinLabCUMC/dissectionReader (accessed on 23 February 2022). Colony sizes from technical replicates were averaged and the standard deviation was determined (Appendix A). The average sizes were then converted into individual sets of Z-scores for each plate in the library, based on the average colony size for each plate (this allowed for comparisons between pRS426 and pRS426-Aly1 or -Aly2 colony sizes). The average colony sizes for each gene deletion strain containing pRS426 were subtracted from the colony size for that gene deletion when over-expressing Aly1 or Aly2 to produce the ‘change from vector’ or ∆V value (Appendix A). These ∆V values were then converted to Z-scores for each plasmid transformant, growth condition, and plate within the library. A Z-score cutoff of +/−1.2 was arbitrarily used to identify gene deletion candidates for further study. To be clear, there are likely other candidates from these screens that significantly impact Aly1 and/or Aly2 function; however, in the interest of having a manageable target set, we chose this cutoff for candidates. From the conditions used (SC-Ura-, 0.8M and 1.5M NaCl, or rapamycin), the Aly1 and Aly2 candidates were combined into a single aggregate candidate list. Gene deletions that altered Aly1- or Aly2-dependent growth substantially in 2 of the 3 medium conditions listed were used to generate the secondary screening candidates list. We initially chose 62 candidates here, however colony PCR used to verify the location of the KANMX4 cassette in the *cla4*∆ and *hog1*∆ isolates from our library revealed that these mutations were not correct and so they were discarded from the screen. These results were in contrast with many other colony PCR validations for strains used in this study. The 60 candidates were assessed for their ability to alter phenotypes using serial dilution growth assays (Appendix A), and, from this secondary screen, 18 candidates were chosen for further study. For these 18 final candidates, we assessed their ability to alter Aly1/Aly2-associated phenotypes (serial dilution growth assays), abundance (immunoblotting and microscopy), electrophoretic mobility (immunoblotting), or localization (microscopy), using the methods described herein.

### 2.5. Yeast Protein Extraction, CIP Treatments, and Immunblot Analyses

Whole-cell extracts of yeast proteins were prepared using the trichloroacetic acid (TCA) extraction method as previously described [58] and as modified from [59]. In brief, cells were grown in SC medium to mid-exponential log phase at 30 °C (A_600_ = 0.6–1.0) and an equal density of cells was harvested by centrifugation. Cell pellets were flash-frozen in liquid nitrogen and stored at −80 °C until processing. Cells were lysed using sodium hydroxide and proteins were precipitated using 50% TCA. Precipitated proteins were solubilized in SDS/Urea sample buffer [8 M Urea, 200 mM Tris-HCl (pH 6.8), 0.1mM EDTA (pH 8), 100 mM DTT, Tris 100 mM (not pH adjusted)] and heated to 37 °C for 15 min. In some instances (Figure 5C,D), 15 μL of cell lysate was subsequently treated for 1 h at 37 °C with 40 units of Quick calf intestinal alkaline phosphatase (CIP, New England Biolabs, Ipswitch, MA, USA) as per the manufacturer’s recommendations, or mock-treated in CIP buffer without enzyme. Samples were then precipitated using 50% TCA and solubilized in SDS/Urea sample buffer as above. Proteins were resolved by SDS-PAGE and identified by immunoblotting with a mouse anti-green fluorescent protein (GFP) antibody (Santa Cruz Biotechnology, Santa Cruz, CA, USA) to detect GFP fused α-arrestins. As a protein-loading control, immunoblot membranes were stained after transfer with Revert^TM^ (Li-Cor Biosciences, Lincoln, NE, USA) total protein stain and were detected using the Odyssey^TM^ FC infrared imaging system (Li-Cor Biosciences). Anti-mouse, anti-rat, or anti-rabbit secondary antibodies, conjugated to IRDye-800 or IRDye-680 (Li-Cor Biosciences), were used to detect primary antibodies on the Odyssey^TM^ FC infrared imaging system (Li-Cor Biosciences). Quantification of band intensities on immunoblot scans was performed using Image J software (NIH, Bethesda, MD, USA). In brief, bands were boxed, and the mean pixel intensity measured. Background fluorescence was subtracted from each measurement and band pixel intensities were compared across replicates.

### 2.6. Protein Stability Assays

The stability of GFP-tagged α-arrestins expressed from the pRS415-*TEF1* promoter plasmids (see Appendix A) was assessed by immunoblotting. Cells were grown to mid-exponential growth phase (A_600_ 0.8–1.0) at 30 °C and cells were treated with 0.1 mg/mL CHX (Gold Bio, St. Louis, MD, USA) to block new protein synthesis; equal densities of cells were harvested over time. Cell pellets were treated as indicated above in Section 2.5, and TCA whole-cell protein extracts were made, resolved by SDS-PAGE, and the GFP-tagged proteins were detected by immunoblotting. To determine the impact of proteasomal inhibition on α-arrestin protein stability, similar assays were performed in cells where the multi-drug resistance pump, Pdr5, was deleted. Cells were then treated with 100 μM MG132, a proteasomal inhibitor (APExBIO, Houston, TX, USA), for 1 h prior to the addition of 0.1 mg/mL CHX. Once again, equal densities of cells were harvested at time points post-treatment with MG132 and cycloheximide. TCA extracts were generated, proteins were resolved by SDS-PAGE and identified via immunoblotting. Quantification of proteins on membrane blots was performed using Image J software, as described in Section 2.5 above.

### 2.7. Fluorescence Microscopy

Fluorescent protein localization was assessed using high-content confocal microscopy. For high-content confocal microscopy, which was done to assess Aly1-, Aly2-, Aly1^PPXYless^, or Aly2^PPXYless^-GFP or Git1-GFP localization across kinase and phosphatase gene deletions, we grew cells overnight in SC medium in either 96-well dishes or culture tubes with orbital shaking or on a rotating drum, respectively, at 30 °C. We then diluted cells and regrew them in 96-well format for 4 h with shaking on a 30 °C platform shaker. Cells were then inoculated to low density (~A_600_ = 0.15) into Cell Carrier 96-ultra microwell plates (Perkin Elmer, Waltham, MA), where each well had been treated with 50 μL of 0.2 μg/μL concanavalin A (MP Biomedicals, Solon, OH, USA) and the bottom of the optical plastic surface that contacts the objective had been coated with RainEx (to allow water to glide along with the objective during imaging). Each well contained 15 μM Cell Tracker Blue CMAC (7-amino-4-chloromethylcoumarin) dye (Life Technologies, Carlsbad, CA, USA) and, for Git1 localization, 10 μM trypan blue (Gibco, Dublin, Ireland), to mark the vacuoles and the cell wall/cell surface, respectively. Cells were imaged using a Nikon Eclipse Ti2-E A1R inverted microscope outfitted with a water immersion 40× LWD objective (NA 1.15), and images were detected using GaAsP or multi-alkali photomultiplier tube detectors. Acquisition was controlled using NIS-Elements software (Nikon) and all images within an experiment were captured using identical settings. Images were cropped and adjusted evenly using NIS-Elements.

### 2.8. Image Quantification and Statistical Analyses

Image quantification was done using the Nikon NIS-Elements, NIS.*ai* (Artificial Intelligence) and Nikon General Analysis 3 (GA3) software package. For quantification of Aly1-, Aly2-, Aly1^PPXYless^, or Aly2^PPXYless^-GFP signal, the NIS.*ai* software was trained on a ground truth set of samples, where cells had been segmented using the images acquired in the DIC channel. Next, the NIS.*ai* software performed iterative training until it achieved a training loss threshold of <0.02, which is indicative of a high degree of agreement between the initial ground truth provided and the output generated by the NIS.*ai* software. Then, fields of images captured via the high-content imaging were processed, so that the cells in a field of view were segmented using the DIC. Any partial cells at the edges of the image were removed. Then, the mean fluorescence intensity for each cell was defined in the GFP channel. Data derived from these kinds of analyses are presented in Figures 3, 4, 6, 9 and Appendix A.

To measure the cell surface fluorescence in comparison to the vacuolar fluorescence for Git1-GFP, a transporter found at the cell surface whose trafficking to the vacuole is regulated by Aly1 and Aly2, we relied on the fiducial markers of trypan blue (TB), which marks the cell wall, and CMAC, which marks the vacuoles. We then trained the NIS.*ai* software to identify the yeast cell surface using the TB-stained cells. TB stains the cell wall, and its fluorescence can be captured using 561 nm laser excitation and 595–650 nm emission. However, it should be noted that TB maximal fluorescence is at 620 nm excitation and 627–720 nm detection, where it effectively marks bud scars and the cell surface. For our approach, it was better to use a very modest portion of fluorescent signal, captured with the 561 nm excitation, so that bud scars were not observed and only faint cell surface fluorescence could be seen. While TB marks the cell wall, the cell wall cannot be distinguished from the plasma membrane at this resolution by fluorescence microscopy, therefore this serves as a marker for the cell surface in our machine learning software [60]. The NIS.*ai* software was trained using a manually defined ‘ground truth’ set of cell surfaces until it achieved a training loss threshold of <0.02, indicating strong agreement between the initial ground truth provided and the output generated by the NIS.*ai* software. In parallel with this, the CMAC staining was used to define the vacuole. A General Analyses 3 (Nikon) workflow was built in Nikon NIS-Elements software to pair the cell surface, as defined by TB staining with the internal vacuolar mask generated by the CMAC staining. This combined mask was used to measure the GFP fluorescence intensity at the cell surface and in the vacuole for all the cells in at least 6 fields of view for Git1-GFP (as shown in Figure 10b).

Fluorescent quantification was then assessed statistically using Prism (GraphPad Software, San Diego, CA, USA). Unless otherwise indicated, we performed the Kruskal–Wallis statistical test with Dunn’s post hoc correction for multiple comparisons. In all cases, significant p-values from these tests are represented as: *, *p* value < 0.1; ** *p* value < 0.01; ***, *p* value < 0.001; ns, *p* value > 0.1. In some instances where multiple comparisons are made, the † symbol may additionally be used in place of the * with the same *p* value meanings, though indicating comparisons to a different reference sample.

### 2.9. RNA Extraction and Relative Quantitative Reverse Transcription Polymerase Chain Reaction (qRT-PCR) Analysis

RNA was extracted from wild-type, *sit4*∆, *npr1*∆, *tor1*∆, *sit4*∆ *npr1*∆, *pep4*∆, or *sit4*∆ *pep4*∆ cells expressing one of three different plasmids: (1) pRS415-*TEF1pr*-GFP, (2) pRS415-*TEF1pr*-Aly1-GFP, or (3) pRS415-*TEF1pr*-Aly2-GFP using a hot phenol/chloroform extraction method, as described in [61]. In brief, 6 mL of midlog phase cells were harvested (OD_600_ = 1.0), washed in water, and flash-frozen in liquid nitrogen. Cell pellets were resuspended in AE buffer (50 mM sodium acetate [pH 5.3], 10 mM EDTA), and SDS was added to a final concentration of 1% (*w*/*v*). This mixture was vortexed, an equal volume of buffer-saturated phenol was added, and it was then incubated at 65 °C for 4 min. The mixture was then rapidly cooled in a dry ice/ethanol bath for 1 min, and microfuge tubes spun at 14,000 RPM for 8 min to separate phases. The top, aqueous layer was transferred to a new microfuge tube, and an equal volume of buffer-saturated phenol/chloroform was mixed into the sample by vortexing before the phases were separated by repeating the spin. The top, aqueous layer was again transferred to a new microfuge tube, and the RNA was precipitated using sodium acetate/ethanol, incubation at −20 °C overnight, and microcentrifugation at 14,000 RPM for 15 min at 4 °C. The supernatant was removed, and RNA pellets were washed with 80% ethanol, followed by microcentrifugation at 14,000 RPM for 15 mins at 4 °C. The supernatant was removed and RNA pellets were dried for 10 min. RNA pellets were resuspended in 75 μL of RNase-free water and the concentration was determined using a Nanodrop One Microvolume UV-Vis spectrophotometer (ThermoFisher Sci., Waltham, MA, USA); the quality of the RNA was determined by resolving 1 μg on a 1% agarose gel containing bleach.

For each sample, 1.5 µg of total RNA was converted to cDNA using a High Capacity RNA-to-cDNA kit (Applied Biosystems^TM^, Waltham, MA, USA). The cDNA was then diluted 5-fold, and 0.5 µL was used in each real-time PCR reaction. Primers used for the PCR were: (1) *GFP* forward (5′CATTACCTGTCCACACAATCT3′) and *GFP* reverse (5′ATCCATGCCATGTGTAATCC3′) to detect all transcripts expressing GFP, (2) *TEF1* forward (5′CTCAAGCTGACTGTGCTATC3′) and *TEF1* reverse (5′CAAGGTGAAAGCCAACAAAG3′) to detect the expression of chromosomal *TEF1*, (3) YEP3 forward (5′CCAGCAATCCATTAAGGTTC3′) and *YEP3* reverse (5′CAGCGGTCTTCTTGTCCTTG3′) to detect expression of chromosomal *YEP3* as a control for cDNA abundance, and (4) *ADH1* forward (5′ATCTTCTACGAATCCCACGG3′) and *ADH1* reverse (5′CCACCGACTAATGGTAGCTT3′). PowerSYBR^®^ Green PCR Master Mix (Applied Biosystems^TM^, Waltham, MA, USA) was used for real-time PCR reactions in a 96-well PCR plate (Millipore Sigma, Burlington, MA, USA). Two replicate wells were assigned for each unique cDNA-primer combination, and cDNAs were prepared from two biological replicates for a total of 4 replicates (2 biological replicates with 2 technical replicates on each) for every cDNA-primer pairing. Platemax^®^ UltraClear sealing films (Millipore Sigma, Burlington, MA) were used to seal the tops of PCR plates. Real-time PCR was performed on a 7900HT Fast Real-Time PCR System (Applied Biosystems^TM^, Waltham, MA, USA) using the default program and the disassociation curve stage. The results were analyzed using the 2^−∆∆CT^ method [62]. Relative quantification (RQ) values to the WT were normalized to the *YEP3* gene as a control for expression. For each set of samples (pRS415-*TEF1pr*-GFP, pRS415-*TEF1pr*-Aly1-GFP, pRS415-*TEF1pr*-Aly2-GFP), expression in mutant strains is presented as relative to expression in wild-type cells. qRT-PCR results are presented as the mean of four replicates ± S.D. from two independent biological experiments. We performed the Kruskal–Wallis statistical test with Dunn’s post hoc correction for multiple comparisons. In all cases, significant *p*-values from these tests are represented as: *, *p* value < 0.1; ** *p* value < 0.01; ns, *p* value > 0.1.

## 3. Results

### 3.1. Identifying Kinases and Phosphatases that Influence Aly1- or Aly2-Mediated Growth in Response to Rapamycin or High-Salt Stress

We sought to identify the kinases and phosphatases controlling α-arrestins Aly1 and Aly2—which have 40 and 87 phosphorylation sites, respectively—in response to rapamycin, which inhibits TORC1 and mimics nutrient deprivation, or growth when in the presence of high salt/osmotic stress. We chose these conditions because: (i) we found that Aly1 or Aly2 over-expression conferred resistance to the Tor1 inhibitor, rapamycin (Figure 1a, [5]), and (ii) several of the membrane proteins controlled by Alys, including Dip5 and Ena1, are internalized in salt stress and Aly-regulators (i.e., the protein phosphatase CN), are activated in high salt (Figure 1a, [42,43,46]). To identify phospho-regulators, we built the KinDel library, a deletion collection sub-library containing all non-essential protein kinases or phosphatases. Importantly, we performed these screens as part of the Introduction to Molecular Genetics laboratory course (BIOSC 0352) at the University of Pittsburgh (Spring of 2020 and 2021). This KinDel screen formed the foundation of a course-based undergraduate research experience (CURE) for 48 undergraduate students, many of whom contributed to the initial observations reported herein. To initiate our screen, we introduced plasmids over-expressing Aly1, Aly2, or a vector control into each KinDel strain, and looked for synthetic genetic interactions between the gene deletions and Aly over-expression on rapamycin or high salt. We measured colony sizes and compared those expressing α-arrestins to vector-containing colonies (see Section 2) and examined the distribution of colony sizes for each condition. Candidates that deviated from the mean colony size (Z-score of > or <1.25) in a condition and differed in growth from the vector-containing control (∆V measure described in Section 2; Appendix A) were selected. This latter comparison ensured that gene deletions causing sensitivity/resistance to rapamycin or salt were not considered further unless they also altered Aly-dependent phenotypes.

From this screen, we identified the genes as indicated in Appendix A (for added detail, please see Appendix A), and candidates are summarized in Figure 1b,c. Surprisingly, 83 and 95 gene deletions (44–51% of the KinDel library) altered Aly1- and Aly2-mediated phenotypes, respectively. We performed extensive secondary screening to refine and validate our candidates. As paralogs, Aly1 and Aly2 share many overlapping functions, therefore we chose to pursue the 60 genes identified in two or more screens. As a more sensitive measure of Aly-induced phenotypes, we compared serial dilution growth assays for each of these 60 gene deletion strains transformed with plasmids over-expressing Aly1 or Aly2 or an empty vector (note that Alys were expressed from their own promoters on high-copy plasmids) (Appendix A). All the gene deletions tested influenced Aly1- and/or Aly2-mediated growth in at least one growth condition, with the most dramatic phenotypes on rapamycin. This suggests that a broad phospho-regulatory web influences Aly function. These gene deletion candidates tended to reduce Aly-mediated resistance to rapamycin, suggesting that these genes normally improve Aly function. Focusing on the rapamycin phenotypes, we restricted subsequent experiments to the 18 genes listed in Table 1.

The gene deletion examined is listed on the left and analyses associated with Aly1 are in blue columns, while Aly2 is in green columns. Data are summarized from the serial dilution spot assays presented in Figure 2 (NaCl and Rapa columns and concentrations are those indicated in Figure 2), and fluorescence microscopy and immunoblotting are presented in Figure 3 and Figure 4 (Protein levels and Mobility shift columns).

### 3.2. Some of the Kinases and Phosphatases that Impact Aly1- Or Aly2-Mediated Growth Phenotypes Alter Aly Protein Abundance

To avoid gene deletion candidates that altered Aly-mediated phenotypes simply by modifying Aly expression from their endogenous promoters, which is a known mode of regulating other α-arrestins (i.e., ECM21/*ART2* and *CSR2*/*ART8*) [9,34]), in follow up studies, the Alys were expressed from the *TEF1* promoter and fused to GFP (pRS415-*TEF1pr*-Alys-GFP in Appendix A). Expression of Aly1 or Aly2 from the robust *TEF1* promoter improved Aly-induced resistance to rapamycin (Figure 2A), suggesting that the Aly-linked rapamycin resistance is dosage-dependent, since the *TEF1* promoter is much more highly expressed than the *ALY*-promoters. Over 60% of the genes tested altered Aly-mediated growth on rapamycin and >50% altered Aly-dependent growth on high salt, though these latter candidates were not further pursued (Figure 2B, Table 1). Interestingly, in many cases, the rapamycin phenotype associated with *ALY* over-expression was tied to the interaction of Alys with Rsp5, as mutants of Aly1 and Aly2 lacking their respective ^L^/_P_PXY motifs—referred to as Aly1^PPXYless^ or Aly2^PPXYless^— grew similar to the vector control (Figure 2B). Aly^PPXYless^ mutants fail to bind Rsp5 and have impaired endocytic function [10,11,55]. Thus, Alys’ trafficking functions are typically important for their response to rapamycin and/or salt stress, with some exceptions. In general, the dependence upon the PPXY motif was more obvious for rapamycin-dependent growth, with *fus3*∆, *gip2*∆, *kin82*∆, *ptc4*∆, *rck2*∆, *sip2*∆, *spo7*∆, *ste7*∆, *tip41*∆, and *ymr1*∆ showing a clear requirement for the PPXY motif, than it was for growth on high salt. In contrast, some gene deletions were sensitive to rapamycin themselves, and caused a near complete loss of Aly-dependent growth on rapamycin (i.e., *ctk1*∆, *nem1*∆, *slt2*∆, *ste20*∆, *tor1*∆, and *yvh1*∆), which makes it impossible to assess the activity of the PPXYless mutants in these strains. In rare instances, there are Aly1- or Aly2-specific phenotypes, the most notable of which was in *sit4*∆ cells, where over-expression of Aly1 increased sensitivity to rapamycin, while over-expression of Aly2 did little to alter the *sit4*∆ rapamycin resistance phenotype (Figure 2B). Unlike the findings on rapamycin, in high salt growth, Aly2 had little effect even in WT cells, while Aly1 caused sensitivity. It is therefore not surprising that there are several deletion mutants where Aly2 has no impact on growth in high salt, while Aly1 causes sensitivity (i.e., *kin82*∆, *ptc4*∆, *rck2*∆, *slt2*∆, *ste7*∆, *tip41*∆, or *tor1*∆). Thus, for Aly2, it is difficult to know if the PPXY motif impacts Aly2 function on high-salt medium, since there is no activity for wild-type Aly2. However, the sensitivity to high salt conferred by Aly1 always required the PPXY motif. The most striking change in high salt occurred in *sip2*∆ cells, which disrupts one of the β-subunits that regulates Snf1. In this background, both Alys gave robust resistance to high salt that was completely dependent upon their PPXY motifs (Figure 2B), which is an exciting finding to pursue in future work.

We further examined the abundance and electrophoretic mobility of GFP-tagged Aly1, Aly2, or their respective PPXYless mutants in the 18 gene deletion candidates using high-content fluorescence microscopy and immunoblotting (Figure 3 and Figure 4 and summarized in Table 1). More than 70% of the gene deletions tested (14 and 13, for Aly1 and Aly2, respectively) altered Aly protein abundance relative to the WT control (Figure 3 and Figure 4 and Table 1). Aly1 and Aly2, as with other α-arrestins, are soluble, cytoplasmic proteins that can be transiently recruited to membranes [3,5,8]. High-content microscopy and automated image quantification (using NIS.*ai* software, see Section 2), allowed us to assess the fluorescence distribution across the population, measuring 100s to 1000s of cells (Figure 3A,B and Figure 4A,B and Appendix A). We found that Alys predominantly localized to the cytoplasm, with some limited fluorescence in the nucleus, and were largely vacuole-excluded. Occasional puncta were observed for Aly-GFP that may correspond to the previously reported Golgi or endosomal localization for Alys [5]. While Aly-GFP abundance was significantly changed in many gene deletion strains, we did not see dramatic redistributions of Alys to new subcellular locations (Figure 3A,B and Figure 4A,B).

When we examined the abundance of free GFP expressed from the *TEF1* promoter in these same gene deletion strains (Appendix A), some had reduced GFP fluorescence. This made us wonder if the diminished fluorescence of Aly-GFP reflected changes in *TEF1* promoter expression in these gene deletion backgrounds. However, upon careful examination of transcript abundance using qRT-PCR targeted to the GFP portion of *ALY1*-*GFP*, *ALY2*-*GFP*, or free *GFP* transcripts, we rarely saw any significant changes in transcript abundance. The one notable exception to this was Aly1-GFP transcript levels in *sit4*∆ cells, which were reduced by ~two-fold, relative to WT cells (Appendix A). In several gene deletion candidates, we also monitored changes in the expression of the chromosomal *TEF1* gene, as this same promoter drives Aly plasmid constructs’ expression. Here, again, most modest changes in transcript abundance in the gene deletion strains were not significantly different from the WT control (Appendix A). Taken together, these findings support the idea that the observed Aly or GFP protein abundance changes are likely due to alterations in protein stability. However, for Aly1-GFP, the two-fold drop in expression in *sit4*∆ cells likely also contributed to the reduced fluorescence, and this is considered further below. Immunoblotting of Aly-GFP proteins extracted from these 18 gene deletion strains confirmed alterations in protein abundance observed by fluorescence microscopy and, in a few instances, we saw altered Aly electrophoretic mobility (Figure 3C and Figure 4C; Table 1), which could indicate changes in phosphorylation.

Excitingly, from our screen, several members of two signaling pathways influenced Aly function: (1) Ste20, Ste7, and Fus3 kinases, which are components of the GPCR-controlled, pheromone-responsive mating pathway in yeast [63,64], and (2) Tor1, Sit4, and Tip41, which are components of the highly conserved TORC1 signaling network that regulates response to nutrient supply [47,49,50,51,52]. While the mating pathway regulation of Alys warrants closer study, especially since Aly1 and Aly2 control the trafficking of the GPCR Ste3 that heads this pathway in *MAT***a** cells [10,11], we chose instead to focus on Sit4 and components of the TORC1 signaling network, as cells lacking *SIT4* had the most dramatic impact on Alys. In *sit4*∆ cells, which themselves are rapamycin resistant, Aly1 over-expression increased rapamycin sensitivity, while the growth of Aly2 over-expressing cells was comparable to the vector control (Figure 2B). Aly protein abundance was reduced, and Aly proteins were slower migrating on immunoblots of extracts from *sit4*∆ cells (Figure 3,Figure 4 and Figure 5B).

### 3.3. Sit4 and Npr1 Regulate Aly1 and Aly2 Phosphorylation and Abundance

The TORC1-Sit4-Npr1 axis regulates α-arrestin-mediated trafficking [5,8,12,13,37]. The current model for TORC1 control of α-arrestins, based on studies of α-arrestins Bul1 and Bul2, is summarized in Figure 5A. We consider TORC1 regulation of *ECM21*/*ART2* gene expression in Section 4. In brief, when TORC1 is active, Sit4-Tap42 is bound to this complex, and Sit4 is unable to dephosphorylate the Npr1 kinase. Npr1 is thus maintained in a phosphorylated state by TORC1 and is considered inactive. Under these conditions, α-arrestins may be selectively dephosphorylated by Sit4 [12,13], allowing them to regulate nutrient transporter endocytosis. In contrast, when TORC1 is inhibited by rapamycin, it no longer phosphorylates Npr1. In these conditions, Sit4 is released from TORC1 and directs its phosphatase activity towards the Npr1 kinase, further activating Npr1. In turn, Npr1 phosphorylates α-arrestins and prevents their endocytic role. However, under these conditions Npr1-dependent phosphorylation may stimulate Aly2-mediated recycling of some nutrient permeases, including Gap1, which is needed at the PM under starvation conditions [5]. Aly2 is a direct substrate for the Npr1 kinase, and Aly1 was recently identified as having Npr1-dependent phosphorylation sites [5,37]. Regulation of Alys by Sit4 has not been previously described.

To define Sit4 regulation of Alys, we more carefully examined the abundance and mobility of Alys and their PPXYless mutants in *sit4*∆ cells (Figure 5B). All forms of Aly have reduced electrophoretic mobility and lower protein abundance in *sit4*∆ cell extracts compared to WT cells (Figure 5B). The electrophoretic mobilities of Aly1 and Aly2 in extracts from *sit4*∆ cells were similar to those from WT cells upon CIP treatment; Alys are hyperphosphorylated in the absence of Sit4 (Figure 5C,D). Since Sit4 and Npr1 co-operate to regulate α-arrestins [12,13], we looked more closely at Npr1 regulation of Alys. It should be noted that, while *npr1*∆ cells were in our initial screen, loss of Npr1 only significantly altered Aly1-mediated growth on rapamycin, and so was not part of our secondary screening.

No obvious electrophoretic mobility changes were observed for the Alys extracted from *npr1*∆ cells, which is not surprising, given that Npr1 is largely inactive in the growth conditions used. However, Aly electrophoretic mobility was similar to that of WT cells in in *sit4*∆ *npr1*∆ cells (Figure 5C,D). We propose that Sit4 prevents Npr1-dependent hyperphosphorylation of Alys in cells grown in the presence of robust amino acids and nitrogen.

The localization of GFP-tagged Alys in cells lacking *TOR1*, *SIT4*, *NPR1*, or both *SIT4* and *NPR1*, was unchanged compared to WT cells (Figure 6a–d). However, Aly2-GFP had significantly lower fluorescence in the *sit4*∆ cells compared to *npr1*∆ or *tor1*∆ cells, each of which modestly reduced fluorescence compared to WT cells (Figure 6a–d). Remarkably, *sit4*∆ *npr1*∆ cells significantly restored Aly2-GFP abundance in comparison to *sit4*∆ cells, and this increase was not due to the elevated transcription of the *TEF1pr-ALY2*-GFP gene (Appendix A). Similar trends to those described for the microscopy were observed for Aly2- or Aly2^PPXYless^-GFP when examined by immunoblotting (Figure 6g,h).

However, unlike Aly2, we often found two distinct subpopulations within our *sit4*∆ *npr1*∆ cultures expressing Aly1-GFP (Appendix A shows distributions for 10 replicate experiments). The first population had restored Aly1-GFP abundance in *sit4*∆ *npr1*∆ cells to near wild-type levels (42% of cells), while the second population had almost no detectible Aly1-GFP (58% of cells) (Figure 6a,b, compare Pop. 1 to Pop. 2; Appendix A). In addition, *sit4*∆ *npr1*∆ cells transformed with the Aly1-GFP expressing plasmid produced a slow- and fast-growing population, as evidenced by a mix of small and large colonies on plates. When regrown, either of these two colony types gave rise to mixed cell populations with both high and low levels of Aly1-GFP fluorescence (Figure 6a,b and Appendix A). Unlike Aly2, *TEF1pr*-*ALY1*-GFP transcript abundance is reduced in *sit4*∆ cells by ~two-fold, and while transcript levels were significantly restored in the *sit4*∆ *npr1*∆ cells, there was again two distinct subpopulations (Appendix A). It is important to note that these transcript abundance changes were not observed for the endogenous *TEF1* gene in *sit4*∆ or *sit4*∆ *npr1*∆ cells (Appendix A). This latter finding suggests that the reduced *ALY1* transcript abundance may be due to altered *ALY1* transcript stability in *sit4*∆ cells, rather than altered expression from the *TEF1* promoter, however we do not pursue the mechanism for this regulation further herein. For the most part, the results of the immunoblot analyses for Aly1- or Aly1^PPXYless^-GFP extracted from cells lacking Tor1, Sit4, Npr1, or Sit4 and Npr1 (Figure 6e,f) reflected the results obtained by microscopy. However, here, we again observed variability in results for Aly1. In no less than seven replicate experiments, we found that, while Aly1-GFP electrophoretic mobility was fully restored in *sit4*∆ *npr1*∆ cells, Aly1 protein abundance only improved sometimes (compare Figure 5C f). The mixed populations observed for Aly1-GFP fluorescence likely explain this variability. Using microscopy, we can carefully characterize the fluorescence distributions for large populations of individual cells (Figure 6a,b), while, in an immunoblot, we read only a single aggregate value for Aly1-GFP protein for the entire population.

The amount of Aly1-GFP in any single protein extract is dependent upon the number of *sit4*∆ *npr1*∆ cells within the population that have reduced Aly1-GFP abundance rather than restored it, and so the amount of Aly1-GFP observed will vary (compare Figure 5C and Figure 6f). Indeed, through microscopy, we found that Aly2-GFP fluorescence did not vary in the *sit4*∆ *npr1*∆ cells across replicates, and that protein levels were consistently restored in our immunoblot analyses. In contrast, Aly1-GFP fluorescence varied significantly between *sit4*∆ *npr1*∆ replicate experiments, therefore it is to be expected that this same variability is reflected in the immunoblots (Appendix A). It is for this reason, among others, that we favor the more nuanced readout from the fluorescence microscopy analyses over immunoblotting when assessing Aly protein abundance. Remarkably, for the Aly1^PPXYless^-GFP protein assessed in four replicate immunoblots (representative blot in Figure 6f), we did not observe the same variability in the *sit4*∆ *npr1*∆ cells. In contrast to Aly1-GFP, Aly1^PPXYless^-GFP was always stabilized and dephosphorylated in the *sit4*∆ *npr1*∆. This suggests that the secondary factor promoting Aly1 instability in the *sit4*∆ *npr1*∆ cells is at least partially dependent upon Aly1′s interaction with the Rsp5 Ub ligase.

Based on these collective observations, we posit that over-expressing functional Aly1 is toxic to *sit4*∆ *npr1*∆ cells and, as a result, there are secondary genetic mechanisms that reduce *ALY1* transcript abundance and protein levels to permit more robust growth. A similar effect was observed with *sit4*∆ *pep4*∆ cells (see results below; Figure 9 and Appendix A), wherein *ALY1* transcript and protein abundances were stabilized in only a subpopulation of cells. From these data, we found that, in the absence of Sit4, the Npr1 kinase is required for hyperphosphorylation and destabilization of Alys. For Aly1, a secondary mechanism of destabilization exists, giving rise to these mixed populations in the *sit4*∆ *npr1*∆ cultures, while, for Aly2, this is the predominant pathway. Sit4 phosphatase may be directly dephosphorylating the Alys to counter Npr1 activity, or perhaps Npr1 phosphorylation of Alys is increased when Sit4 is lost. Indeed, under the conditions employed, we would anticipate that Npr1 kinase activity would be limited due to phospho-inhibition by TORC1. Further, Npr1 is hyperphosphorylated and thought to be inactive in the absence of Sit4 [47]. For these reasons, we favor a model wherein the loss of the Sit4-mediated dephosphorylation of Alys results in a failure to counterbalance even modest Npr1-controlled phosphorylation, which might be expected under these conditions. Over time, this could lead to Aly hyperphosphorylation.

### 3.4. TORC1 Inhbition or Loss of Sit4 Induces Aly1 and Aly2 Instability and Defective Vacuole Function Restores Aly Levels

The TORC1 pathway is not known to control the stability of α-arrestins. To define the contribution of the TORC1 pathway to Aly stability, we examined the abundance of GFP-tagged Alys extracted from cells that were either treated with rapamycin, to inhibit TORC1, or CHX, to block protein synthesis, in comparison to untreated cells. These drug treatments are routinely used, but they each have secondary effects that often go unconsidered: rapamycin impedes TORC1 function, which is an activator of protein synthesis, and so this drug indirectly dampens protein expression, and CHX not only blocks protein synthesis, but also activates TORC1 [65,66,67]. Considering these added roles, these data need to be carefully interpreted. Treatment with rapamycin resulted in the most rapid loss of Alys, with nearly all of the Aly protein being lost by 4 h post-treatment (Figure 7A–D). Cycloheximide treatment similarly reduced Aly abundance; however, between 20 and 40% of the initial Aly protein remained at 4 h post-treatment (Figure 7A–D). This is in contrast with untreated cells, where most of the Aly protein remained intact after 4 h, though Aly2 abundance did diminish somewhat, suggesting that, as cells approach saturation, Aly2 levels may decline. Based on these results, we concluded that rapamycin treatment destabilizes Aly1 and Aly2 more effectively than CHX treatment, however both acute treatments result in a loss of Alys, and it is difficult to say if the rapamycin-induced loss is from the acute impairment of TORC1 activity or impaired protein synthesis.

In response to rapamycin, Aly2 and Aly2^PPXYless^ electrophoretic mobility was reduced over time, suggestive of modification prior to degradation (Figure 7C,D). Since this mobility shift occurred in both Aly2 and the PPXYless mutant that cannot be ubiquitinated by Rsp5, we posited that phosphorylation might be the driver for this mobility shift. When TORC1 is inhibited with rapamycin, Npr1 is partially dephosphorylated, though not to the same extent as it is in nitrogen-starved cells [8,12,32,51]. Dephosphorylation of Npr1 is associated with increased Npr1 kinase activity [8,12,32,51]. We reasoned that Npr1 might be required for Aly degradation post-rapamycin treatment, and that Npr1-dependent phosphorylation might be the cause of the observed mobility shift. However, there was no change in the Aly degradation profiles post-rapamycin treatment in WT or *npr1*∆ cells (Figure 7E,F). Further, the mobility shift for Aly2 post-rapamycin treatment was preserved in *npr1*∆ cells. Thus, the rapamycin-induced changes in Alys are not Npr1 dependent (Figure 7E,F).

TORC1 activity directly influences Sit4 function, therefore we next turned our focus towards defining how Alys are degraded in *sit4*∆ cells. We explored the contributions of the proteasome and vacuole to Aly degradation in wild-type and *sit4*∆ cells. First, we impaired proteasome function by treating cells lacking the Pdr5 multidrug-resistant transporter with the proteasome inhibitor MG132 [68].

Cells were incubated in either DMSO or MG132 for 60 min prior to the addition of CHX, and protein abundances were monitored by immunoblotting. Proteasome inhibition with MG132 resulted in a slight increase in Aly1 or Aly1^PPXYless^ prior to CHX addition (i.e., the steady-state abundance of Alys increased when the proteasome was impaired, compare t = 0 lanes in Figure 8A), but the degradation profiles for these proteins post-CHX were not discernibly different (Figure 8A). There was no accumulation of Aly2 or Aly2^PPXYless^ prior to CHX addition (i.e., treatment with MG132 did not elevate steady-state levels, compare t = 0 lanes in Figure 8C), and the rates of turnover appeared similar post-CHX (Figure 8C). Proteasome inhibition did very little to change the degradation of Alys in *sit4*∆ cells (Figure 8B,D). We confirmed that the MG132 treatment was working by using a bona fide proteasome-dependent substrate known as Nbd2* (graciously provided by Jeff Brodsky, Univ. of Pittsburgh) ). Unlike the Alys, we found that Nbd2* was dramatically stabilized in response to MG132 (Figure 8 and Appendix A). Degradation in the proteasome is often preceded by ubiquitination. Therefore, in parallel, we assessed the abundance of Alys in cells lacking E3 ubiquitin ligases and found that Alys were not increased in any of the mutants tested (Appendix A). Interestingly, deletion of the Asi1 ubiquitin ligase, which is associated with protein quality control at the inner nuclear membrane, altered Aly1-GFP electrophoretic mobility and significantly reduced Aly1 abundance (Appendix A), however the impact of this ligase on Aly function was not tested further. Based on these combined findings, the proteasome does not appear to be a significant driver of Aly degradation in these conditions. This contrasts with the proteasome-specific degradation reported for α-arrestin Ldb19/Art1 [33].

We next determined if the vacuole had a role in the degradation of Alys by deleting Pep4, a vacuolar aspartyl protease required for the maturation of most vacuolar proteases [69], from either WT or *sit4*∆ cells. Aly1-GFP, assessed by fluorescence microscopy, was modestly increased in *pep4*∆ cells in comparison to WT cells, but Aly2-GFP showed no such increase (Figure 9A–D). These data are mirrored in the immunoblots where, when we look at the starting abundance of Alys prior to the addition of CHX (compare t = 0 lanes in Figure 9E,F, quantified in Figure 9I), there is little to no difference between Aly abundance in WT and *pep4*∆ cells. However, when we compared Aly-GFP abundance in *sit4*∆ cells to that of *sit4*∆ *pep4*∆ cells, Aly2 was dramatically stabilized (Figure 9B,D), while, for Aly1, we once again observed two populations (Figure 9A,C). In the first population, Aly1 abundance was restored to near WT levels; however, in the second population, very little fluorescence was detected (Figure 9A,C). This variability in Aly1 protein abundance may be partially explained by *ALY1*-GFP transcript levels, as these were significantly higher in *sit4*∆ *pep4*∆ cells than they were in *sit4*∆ cells (Appendix A). As had occurred for Aly1 in *sit4*∆ *npr1*∆ cells, a mix of large and small colonies formed when Aly1-GFP was over-expressed in *sit4*∆ *pep4*∆ cells.

This finding further supports the idea that increased Aly1 protein is toxic and that there is a secondary mechanism available to ensure a reduction in Aly1 transcript or protein levels. As seen in the *sit4*∆ *npr1*∆ immunoblots for Aly1 abundance, Aly1-GFP immunoblots from *sit4*∆ *pep4*∆ cells did not show any increased Aly1 abundance compared to the *sit4*∆ cells alone. Again, we think this result is tied to the fact that the immunoblot is showing a single aggregate value for Aly1-GFP from across the population, while, on a cell-by-cell basis, as seen in the microscopy, the alterations within the population can be observed. Interestingly, expression of the Aly1^PPXYless^ mutant, which lacks the capacity to bind Rsp5, showed no mixed populations in the microscopy and had a robust rise in Aly1^PPXYless^ signal in immunoblots (Appendix A and Figure 9G). These data suggest that the varying Aly1 abundance and altered colony growth are caused by Aly1-Rsp5-associated activities.

In the absence of Pep4, the steady-state abundances of GFP-tagged Alys, for the WT Alys and their respective PPXYless mutants, tended to be slightly higher than the steady-state levels in WT cells, though this effect was not always significant (Figure 9C–F compare t = 0 timepoints, Figure 9I, and Appendix A). In the *sit4*∆ cells, loss of *pep4*∆ significantly increased Aly2, Aly2^PPXYless^, and Aly1^PPXYless^ abundance, but did not significantly increase Aly1 when examined via immunoblotting (Figure 9G,H, compare t = 0 points and Figure 9J). For all the Alys examined, there was a faster-migrating band in the *sit4*∆ *pep4*∆ cells that was not found in the *sit4*∆ extracts (Figure 9G,H, marked in white asterisks). This may indicate that, as with the *sit4*∆ *npr1*∆ cells, the loss of Pep4 has stabilized a pool of the Alys that are not phosphorylated. When we further examined the stability of the Alys in these backgrounds post-CHX treatment, which blocks new protein synthesis, we found that, generally, the Alys were more stable in the *pep4*∆ cells than in the corresponding WT or *sit4*∆ backgrounds, with the notable exception being Aly1 (Figure 9).

One surprising facet of these findings is that the GFP fluorescence does not accumulate in the vacuole in the *sit4*∆ or *sit4*∆ *pep4*∆ cells. If, in *sit4*∆ cells, Aly-GFP is targeted to the vacuole for degradation, and therefore gives rise to reduced Aly protein abundance, we would expect to see free GFP fluorescence accumulate in the vacuole in these cells, since GFP is a stably folded protein that is somewhat resistant to degradation by vacuolar proteases [4,70]. Yet, this was not seen for the *sit4*∆ cells, and instead an overall loss of cytoplasmic fluorescence was observed (Figure 6a,c). We also anticipated that the loss of the vacuole proteases would stabilize the Alys by preventing their vacuolar degradation and thereby lead to an accumulation of fluorescence in the vacuole if this was the route by which the Alys are degraded. However, in no instance do we see the fluorescent signal accumulating in the vacuole; instead, in the *sit4*∆ *pep4*∆ cells, the Aly-GFP fluorescence accumulates in the cytoplasm (Figure 9A,B). Though we do not know why this is the case, we suspect that the loss of the vacuole proteases may be altering nutrient balance in cells (i.e., changing the abundance of free amino acids or other nutrients in the vacuole), thereby altering cellular signaling and the modification/stability of Alys. Changes in cellular signaling could explain why the electrophoretic mobility of Alys is altered in the *sit4*∆ *pep4*∆ cells (Figure 9G,H).

These findings demonstrate that, in *sit4*∆ cells, Aly1 and Aly2 are being degraded in a manner that is partially dependent upon the protease activity of the vacuole. It is worth noting that, upon CHX treatment, we do still see degradation of all forms of Alys in our time course, even in *pep4*∆ cells. This suggests that, when the vacuole is impeded, the proteasome likely now degrades the α-arrestins and so they, like the other substrates that have been described [71], may access both pathways for degradation with vacuolar function, making a substantial contribution.

### 3.5. Sit4 and Npr1 Regulate Aly-Mediated Trafficking of the Git1 Transporter

With our new understanding of Aly-regulation by Sit4 and Npr1, we next explored the impact of this signaling on protein trafficking. We recently showed that Alys control the basal and ligand-induced endocytosis and intracellular sorting of the glycerophosphoinositol transporter Git1 [55]. To determine if the trafficking of Git1-GFP is regulated by TORC1-Sit4-Npr1, we expressed Git1-GFP in cells lacking each of these signaling regulators and assessed Git1 localization and abundance/vacuole proteolysis via microscopy and immunoblotting. As expected, Git1 was localized to the PM and vacuole in WT cells, and this was reflected in the immunoblotting, where there was a balance of full-length Git1-GFP and free GFP, which represents the cleaved GFP that is stably folded and recalcitrant to degradation in the vacuole (Figure 10a–c). Upon the loss of α-arrestins (either *aly1*∆ *aly2*∆ cells or 9Arr∆ cells, which lack 9 of the 14 yeast α-arrestins [4]), Git1 was retained at the PM with little to no vacuolar fluorescence and was better maintained as the full-length Git1-GFP in protein extracts (Figure 10a–c). In the absence of Tor1, we observed a modest retention in Git1 at the PM and a subtly higher PM/vacuole fluorescence ratio, however this subtle change is not reflected in the immunoblotting, and so may not be biologically significant (Figure 10a–c). Remarkably, we saw a significantly higher PM/vacuole ratio in *npr1*∆ cells than in WT cells, suggesting that Npr1 is an important positive regulator of Git1 trafficking to the vacuole (Figure 10a,b). This change in Git1 trafficking to the vacuole is reflected in the immunoblot, where there is less free GFP in the *npr1*∆ (Figure 10c). The effect in *npr1*∆ cells was not as dramatic as the retention at the PM in the absence of α-arrestins and, instead, was intermediate between the WT and 9Arr∆, based on both the imaging analyses and immunoblotting. In *sit4*∆ cells, Git1 is fully retained at the cell surface and has a PM/vacuole ratio comparable to that observed in the 9Arr∆ cells (Figure 10a,b), and almost no free GFP signal is observed via immunoblotting (Figure 10c). This is as we expected, given that the Alys are needed for Git1 internalization and that, in the absence of Sit4, there is very little Aly protein available to stimulate Git1 trafficking (Figure 5B). The additional loss of Npr1 in *sit4*∆ cells restored Git1 trafficking to the vacuole, with the same balance of PM/vacuole localized Git1 and Git1-GFP/free GFP as that of the *npr1*∆ cells (Figure 10a–c). This finding further demonstrates the epistatic relationship between Npr1 and Sit4 for this α-arrestin-regulated activity. In the *sit4*∆ *npr1*∆ cells, Aly2 protein levels are largely restored, and the Alys are once again in a dephosphorylated state (Figure 5C,D), which is likely sufficient to restore the α-arrestin-mediated trafficking of Git1. Together, these findings lead us to propose a new model for α-arrestin regulation by the TORC1-Sit4-Npr1 signaling axis (Figure 10d). Sit4 is required to stabilize the α-arrestins and alter their phosphorylation; Git is retained at the cell surface in *sit4*∆ cells. This loss of Git1 trafficking likely reflects the reduced abundance of Alys, since Git1 is trafficked in an Aly-dependent manner [55]. Finally, the loss of Npr1 in *sit4*∆ cells improved Aly abundance, prevented their hyperphosphorylation, and restored Git1 trafficking to the vacuole. These findings suggest that Npr1 kinase activity is negatively regulated by Sit4. In the absence of Sit4, Npr1 is needed to hyperphosphorylate the α-arrestins. This modification leads to destabilization of the α-arrestins Aly1 and Aly2 and thereby prevents proper Git1 trafficking to the vacuole.

## 4. Discussion

Phospho-regulation of α-arrestins Aly1 and Aly2 is complex, and herein, we identified several kinases and phosphatases that influence Alys stability and/or alter their electrophoretic mobility. Though many exciting regulatory pathways for Alys are suggested from our work, we focus on the interplay between Alys and the TORC1-Sit4-Npr1 signaling pathway. There is a robust literature connecting α-arrestins to TORC1 signaling, and one model finds that, for cells with active TORC1, α-arrestins are dephosphorylated and can mediate endocytosis of nutrient permeases. In contrast, when TORC1 is impaired, by nutrient restriction or rapamycin treatment, α-arrestins are hyperphosphorylated by Npr1, and this impedes their endocytic function. Elegant quantitative mass spectroscopy approaches have defined TORC1- and/or Npr1-dependent phosphorylation sites on several α-arrestins, including Ldb19/Art1, Ecm21/Art2, Bul1, Bul2, as well as Aly1 and Aly2 [8,37]. Our findings lend support to the idea that Aly1 and Aly2 are regulated by TORC1 and Npr1, but we identify Sit4 and other linked regulators, including Tip41 and Slt2 (discussed below), that influence this pathway. Our data suggest that Npr1-mediated phosphorylation of α-arrestins can occur even when TORC1 is active. α-Arrestins Aly1 and Aly2 are hyperphosphorylated in *sit4*∆ cells grown in rich medium, where TORC1 should be active and Npr1 inhibited. Similarly, work from the Andre lab found that α-arrestins Bul1 and Bul2 are hyperphosphorylated in *sit4*∆ cells, irrespective of whether a rich- or poor-quality nitrogen source (which activates or inhibits TORC1, respectively) was used [12]. Unlike the Buls, we find that Alys are unstable in *sit4*∆ cells due to Npr1-regulated hyperphosphorylation, which somehow triggers Aly degradation. Here, the regulation of Aly1 and Aly2 differs slightly, with Sit4 somehow promoting Aly1 transcript stability while Sit4 had no impact on Aly2 transcript abundance. In *sit4*∆ *npr1*∆ or *sit4*∆ *pep4*∆ cells, Aly protein phosphorylation and abundance was similar to that of wild-type cells. Thus, we conclude that Npr1 and the vacuolar proteases are important for the degradation of Alys in *sit4*∆ cells. Surprisingly, in the vacuole protease deficient backgrounds, we did not see Aly-GFP fluorescence accumulating in the vacuole. The alteration in electrophoretic mobility for *sit4*∆ *pep4*∆ cells also leads us to speculate that the loss of the vacuole proteases in this background may be changing cellular signaling to dampen or prevent the Npr1-mediated phosphorylation of the Alys. Since TORC1 is located on the limiting membrane of the vacuole, it could be that altering the free amino acid pool in the vacuole by preventing protease function, as occurs in *pep4*∆ cells, might change TORC1 and downstream signaling.

This will be an interesting idea to pursue in future studies. Finally, in the absence of Sit4, the Aly-regulated Git1 transporter is retained at the cell surface, mirroring what is seen in *aly1*∆ *aly2*∆ or 9Arr∆ cells. Inhibition of Git1 internalization is restored in *sit4*∆ *npr1*∆ cells, where Aly protein levels are also improved. Below, we frame our findings in the larger context of the TORC1-Sit4-Npr1 regulation of other α-arrestins, highlighting conserved and unique regulatory features in comparison to the Alys.

α-Arrestin Ldb19/Art1 is a direct substrate of the Npr1 kinase, and Npr1- and rapamycin-dependent phosphorylation sites have been defined [8]. When TORC1 is impaired via rapamycin treatment, Npr1 phosphorylation is somewhat reduced, though to a lesser extent than the Npr1 dephosphorylation/activation associated with amino acid or nitrogen starvation conditions [8,12,51]. In rapamycin-treated cells, Ldb19/Art1 is hyperphosphorylated in an Npr1-dependent manner [8]. Amino acids in the N-terminus of Art1 are directly modified by Npr1; however, there are C-terminal phosphorylation sites in Art1 that, while not directly phosphorylated by Npr1, rely on the modification of the N-terminal Npr1-regulated sites as ‘priming’ phosphorylation events [8]. These secondary, C-terminal phospho-sites are regulated by an as-yet-unidentified kinase, and reveal an added layer of this regulatory pathway. In response to rapamycin, phosphorylated Art1 cannot stimulate cycloheximide-induced internalization of the arginine permease, Can1 [8]. However, when TORC1 is activated by cycloheximide treatment alone, Art1 is dephosphorylated and recruited to the cell surface. In this way, TORC1-mediated inhibition of Npr1 controls the localization and activity of the α-arrestin. In contrast to this model, however, other research found that Can1 is retained at the PM in *npr1*∆ cells treated with rapamycin [54]. This finding runs counter to the model that Npr1 is impeding Art1 function, as, if this were the case, then Art1 should be hyperactivated and induce endocytosis of Can1 in the *npr1*∆ cells. Unlike what we observed with Aly1 and Aly2, there were no changes in Art1 stability in response to either rapamycin or cycloheximide. In addition, we did not observe any dramatic re-localization of Alys in cells lacking any of these signaling regulators, though perhaps Aly localization post-rapamycin should be monitored in future experiments. The phosphatase that dephosphorylates the Npr1-dependent sites in Art1 remains to be defined. While the protein phosphatase Z isoforms, Ppz1 and Ppz2, are important for Art1 dephosphorylation during methionine-stimulated endocytosis of Mup1, these phosphatases did not dephosphorylate Npr1-dependent phospho-sites in Art1 [26]. To date, the role of the Sit4 phosphatase in Art1 regulation has not been studied.

Studies from the Andre lab extend our understanding of α-arrestin regulation by incorporating Sit4 [12,13,27]. Regulation of Bul1 and Bul2 by Npr1 predates their identification as members of the α-arrestin family of trafficking adaptors [49,72]. Bul regulation by TORC1 is best defined in response to changes in nitrogen source quality. In cells grown in proline, a poor nitrogen source, TORC1 is inactive, and it is thought that the Tap42 protein, which interacts with Sit4 to hold it in complex with TORC1, releases the Sit4 phosphatase. Recent studies have suggested the regulation of Sit4 by Tap42 and TORC1 is more nuanced than previously considered, and Sit4 appears to be active towards some substrates even when bound to Tap42 [52,53]. In line with this model, in proline-grown cells, Sit4 dephosphorylates Npr1 the and loss of TORC1 activity prevents further inhibitory phosphorylation of Npr1 under these conditions, activating the kinase [12,13]. Npr1 then phosphorylates the Buls, which become bound to 14-3-3 proteins, preventing their interaction with Gap1 [12,13]. Under these conditions, Gap1 is needed at the cell surface to help take up amino acids, therefore the inability of Buls to remove Gap1 from the PM aligns well with the cellular physiology [73,74,75]. However, when ammonium (a rich nitrogen source) is added back to proline-starved cells, TORC1 is activated. Instead of inactivating Sit4, TORC1 alters the substrate preference of Sit4 so that it now dephosphorylates the Buls, rather than Npr1. In *sit4*∆ cells, Buls remain hyperphosphorylated, as we similarly observed for Alys in the absence of Sit4. However, unlike the Alys, the Bul proteins appear stable in the *sit4*∆ cells, demonstrating a distinct regulation for the Alys. Dephosphorylated Buls are freed from 14-3-3 proteins, allowing Bul-mediated endocytosis of Gap1 [12,13,27]. We have not determined if 14-3-3 protein binding to Alys is altered in a Sit4-dependent manner. However, we do know that the Alys bind 14-3-3 proteins when cells are grown in a nutrient-rich environment, where TORC1 should be active [28]. It will be interesting to see in future studies if the association of Alys with 14-3-3 proteins is regulated by TORC1.

As an exciting foil to the TORC1 regulation of the Buls, recent studies of Ecm21/Art2 revealed a very different mode of control [9,32]. Powerful quantitative mass spectroscopy identified many new membrane proteins that are regulated by α-arrestins [32]. More specifically, this study showed that endocytic trafficking of the B-vitamin transporter Thi7 is stimulated by α-arrestin Art2 in response to thiamine, Thi7′s substrate, or when TORC1 is inhibited with rapamycin. This is exactly the opposite of the TORC1 regulation observed for the Buls, where impaired activity of TORC1 alleviates repression of Npr1, leading to the phosphorylation of α-arrestins and the retention of Gap1 at the cell surface. In *sit4*∆ cells, Thi7 was retained at the cell surface, irrespective of TORC1 activation, demonstrating once again that α-arrestin function depends on Sit4 [32]. We similarly find that, in the absence of Sit4, the Git1 transporter is retained at the cell surface, consistent with impaired Aly function. However, surprisingly, Thi7 endocytosis still occurred, though somewhat less efficiently, in cells lacking Npr1. This again is in marked contrast to the regulation of Gap1 by Npr1/Buls, as Gap1 is more robustly internalized in *npr1*∆ cells due to a loss of Buls’ phospho-inhibition that activates their endocytic role [12]. Thus, while Npr1 antagonizes the Bul’s ability to control Gap1 internalization, it modestly improves Art2-mediated endocytosis of Thi7. We observe a similar subtle reduction in Git1 trafficking when Npr1 alone is missing, suggesting that Npr1 may both positively and negatively control Aly function, depending on the conditions.

In addition to impeding Art2-mediated endocytosis of Thi7, TORC1 prevents the expression of the *ART2* gene. When cells are grown in nitrogen-starved conditions (inhibiting TORC1), *ART2* expression is upregulated by the Gcn4 transcription activator binding to its promoter [9]. Gcn4 phosphorylation and nuclear activity is controlled by the upstream Gcn2 kinase, which is activated by Sit4 and inhibited by TORC1. Thus, not only can the TORC1-Sit4 signaling pathway regulate α-arrestins via control of their post-translational modifications, it can also alter α-arrestin expression. This is part of the reason why we used the *TEF1* promoter to assess Aly activity, as it avoided potential confounding transcriptional regulation. The regulation of Art2 presents an interesting functional dichotomy for the α-arrestins, with starvation inducing expression and activating the Art2-mediated endocytosis of many nutrient permeases, including Mup1, Can1, and Lyp1 [9]. These same transporters are also regulated by Art1; however, unlike Art2, TORC1 activates Art1-mediated endocytosis [8]. Art1 is thought to regulate endocytosis of these transporters in response to excess ligand to prevent toxic amino acid accumulations, while Art2-controls the removal of these transporters under prolonged starvation, where these specific amino acid transporters are no longer required. It makes sense that, under these same conditions, the Bul-mediated removal of Gap1 is impaired, since this is when Gap1 is needed at the cell surface. It will be interesting in the future to see if all α-arrestin-dependent internalization pathways are equally efficient (i.e., is Art1 as good at internalizing Can1 as Art2?) and to determine if transporters internalized by different α-arrestins follow the same or distinct post-endocytic intracellular trafficking routes. Perhaps, under nitrogen starvation conditions, Art2 is expressed because it is less selective than Art1 and the other α-arrestins. Nitrogen starvation induces a robust endocytosis of many different membrane proteins, and so a less selective α-arrestin could be beneficial under these conditions. Analogously, under glucose starvation conditions, there is a dramatic increase in membrane protein endocytosis and, under these conditions, the α-arrestin Csr2/Art8 is expressed. Csr2 replaces Rod1/Art4 and Rog3/Art7 in regulating glucose transporter endocytosis under these conditions, and perhaps it also promiscuously controls endocytosis of other transporters as well. Added biochemical insights into α-arrestin binding affinities for transporters and their endocytic efficiency would be helpful to the field.

Finally, in addition to the interesting new aspects of Aly regulation by TORC1-Sit4-Npr1, our screen identified Tip41 and Slt2 as related candidates that remain to be pursued. Tap42-interacting protein 41, or Tip41, is an activator of Sit4. When Tap42 is bound to Tip41, Sit4 is released from Tap42, and this likely alters its substrate preferences [12,13,47]. In our screen, we found that Aly1 and Aly2 abundance in *tip41*∆ cells were reduced, similar to what was observed in *sit4*∆ cells. This suggests a model where Tip41 might help promote Sit4-mediated dephosphorylation of α-arrestins, thereby preventing their phospho-dependent degradation. It will be interesting to examine Npr1 phospho-status in the *tip41*∆ cells and *sit4*∆ cells, as earlier studies have suggested that Npr1 is hyperphosphorylated when these regulators are missing, which is typically associated with Npr1 inactivation [47]. Npr1 phosphorylation occurs in degrees, with near-complete dephosphorylation in response to amino acid and nitrogen starvation, but more modest/intermediate phosphorylation upon rapamycin-induced inhibition of TORC1. Since TORC1 is impaired by rapamycin, this latter finding suggests that Npr1 phosphorylation is regulated by additional means. Depending on its degree of phosphorylation, Npr1 may be selectively active towards specific substrates. Another facet of Npr1 regulation that is not considered herein is the phosphorylation of Npr1 by Slt2, the MAP kinase in the cell integrity pathway [37]. Slt2, which is regulated in part by the TORC2 complex [76], can phosphorylate Npr1 to impair its function [37]. We find that, in *slt2*∆ cells, Aly2 but not Aly1 protein levels are reduced. Does Slt2 help maintain Npr1 in a hyperphosphorylated state, preventing phosphorylation of Aly2 and blocking its degradation? The interplay between TORC1 and TORC2 regulation of Slt2 and Npr1 and their combined impact on α-arrestins merits further study.

Many key facets of α-arrestin regulation by TORC1 remain to be explored. While some aspects of this regulation appear conserved for multiple α-arrestins, there are also interesting unique elements for specific α-arrestins. The ability of TORC1 to stimulate—and Npr1 or rapamycin treatment to impede—Art1-, Bul-, or Aly-mediated endocytosis of transporters provides one regulatory paradigm. Yet, our earlier work also showed that Aly2 is a direct substrate of Npr1 and that, under nitrogen starvation conditions, Npr1 phosphorylation of Aly2 is needed to allow Gap1 trafficking to the cell surface [5]. The activation of Art2-mediated endocytosis in response to rapamycin suggests yet another opposing regulatory model to that of Art1, Buls, or Alys. There remain facets of the TORC1-Sit4-Npr1 regulation of α-arrestins that are somewhat confounding. For example, our finding that Sit4 is needed to oppose Npr1 function when cells are grown in a good nutrient supply, where TORC1 should be blocking Npr1 kinase activation, is surprising. It is also in contrast to the observed hyperphosphorylation of Npr1, and presumed inactivation of this kinase, that has been reported for *sit4*∆ cells. The idea that kinase activity can be selectively activated towards specific substrates (i.e., while one condition prevents a kinase from recognizing some substrates, this same condition can selectively promote kinase activity towards other substrates) is an important emerging theme in phospho-regulation. Given our findings and the state of the field, there is undoubtedly more to be gained from further study of TORC1-Sit4-Npr1 regulation of α-arrestins.

## 5. Conclusions

We define new connections between the TORC1-Sit4-Npr1 phospho-regulators and α-arrestins Aly1 and Aly2. Unlike other α-arrestins, which can be controlled by this signaling pathway, the activity of Sit4 is required to maintain Aly stability and function. Aly-degradation and hyperphosphorylation in *sit4*∆ cells require functional Npr1, suggesting that Npr1-mediated phosphorylation of these α-arrestins is a cue for their degradation. Impairing vacuole protease activity in *sit4*∆ cells improves Aly2, but not Aly1, stability.

## Figures and Tables

**Figure 1 biomolecules-12-00533-f001:**
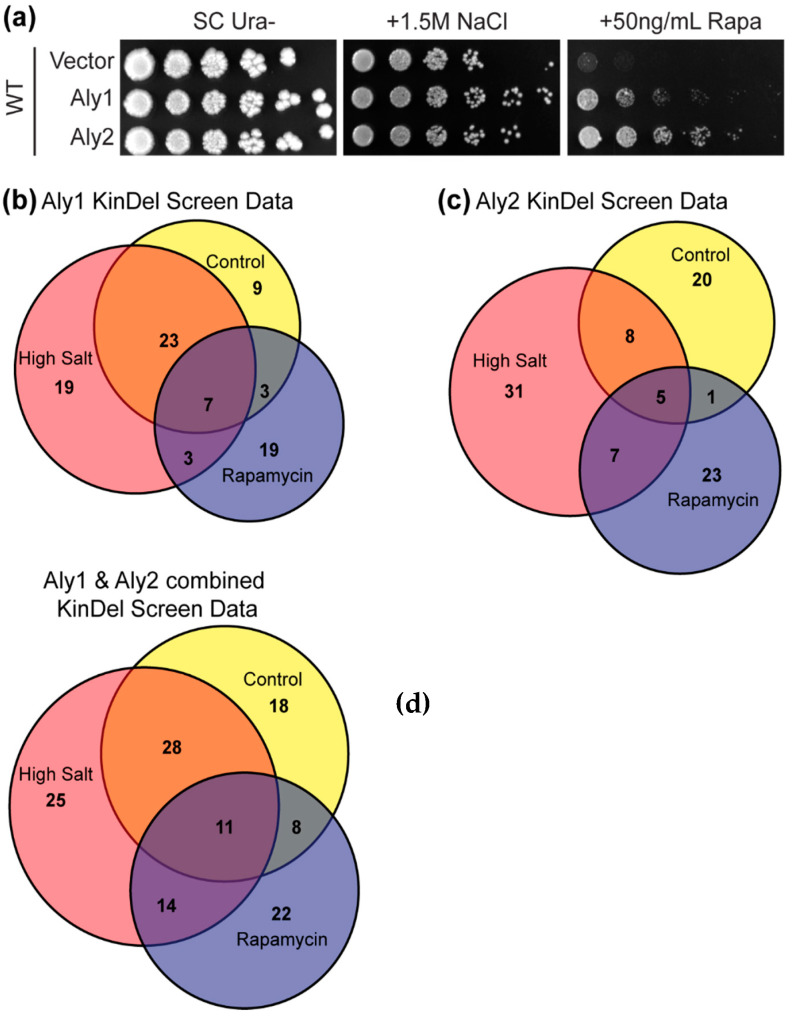
KinDel candidates that influence α-arrestin-dependent phenotypes: (**a**) Growth of serial dilutions of WT cells containing the indicated pRS426-derived plasmids expressing either nothing (vector) or the indicated α-arrestin on SC medium lacking uracil and containing 1.5M NaCl or 50 ng/mL rapamycin. (**b**–**d**) Venn diagrams indicating the number of KinDel library gen deletions found to alter the sensitivity of (**b**) Aly1 or (**c**) Aly2 on control (yellow), high-salt (red), or rapamycin (blue) medium. The sum of the numbers in each circle identifies the number of gene deletions in each class. Panel (**d**) shows the overlapping candidates identified in both the Aly1 and Aly2 screens for each medium condition.

**Figure 2 biomolecules-12-00533-f002:**
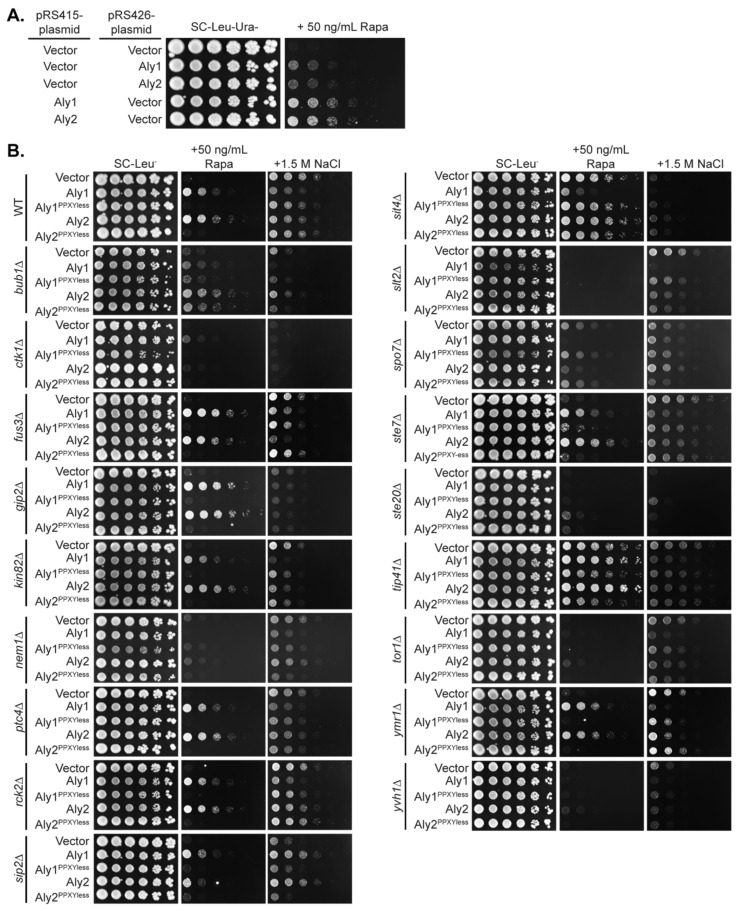
Secondary screening of KinDel candidates for growth phenotypes linked to Aly1- or Aly2-overexpression: (**A**) Serial dilution growth assay to compare the pRS426-*ALY* promoter-expressed α-arrestins to the pRS415-*TEF1* promoter-expressed α-arrestins. Growth of serial dilutions of WT cells containing the indicated pRS426 or pRS415-derived plasmid expressing either nothing (vector) or the indicated α-arrestin. (**B**) Serial dilution growth assays of WT cells or those lacking the gene indicated (from the KinDel library) and containing the indicated pRS415-*TEF1pr*-derived plasmid on the medium indicated.

**Figure 3 biomolecules-12-00533-f003:**
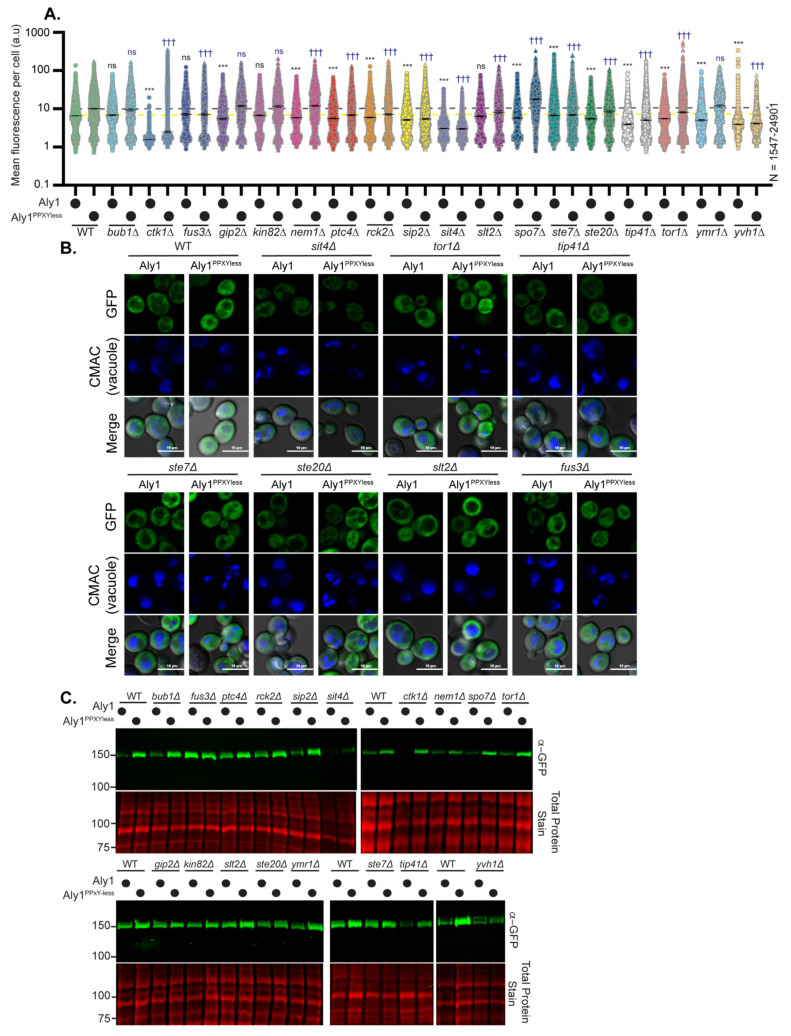
Kinases and phosphatases from the KinDel screen alter the abundance and electrophoretic mobility of Aly1: (**A**) Cells expressing the indicated Aly1 protein fused to GFP (pRS415-*TEF1pr* plasmids) in either WT or the noted gene deletion backgrounds were imaged by high-content confocal microscopy and whole-cell fluorescence of the cells was quantified using NIS.*ai* and Nikon GA3 software. The mean fluorescence intensity from whole-cell measurements (in arbitrary units, au) is plotted for each cell as a circle. The median fluorescence intensity is shown as a black line for each group and the error bars represent the 95% confidence interval. A yellow or black dashed line represents the median fluorescence intensity for Aly1 or Aly1^PPXYless^ expressed in WT cells, respectively. Kruskal–Wallis statistical analysis with Dunn’s post hoc test was performed to compare the fluorescence distributions to the cognate WT. In black asterisks (*) or blue daggers (†), comparisons are made to Aly1 or Aly1^PPXYless^ in WT cells, respectively (ns = not significant; three symbols have a *p*-value < 0.0005). (**B**) A subset of the fluorescent microscopy images acquired for the data presented in (**A**) are shown. CMAC is used to stain the vacuoles (shown in blue). Merge is overlaid with the transmitted light cell image as well as the fluorescence images. (**C**) Whole-cell extracts from the cells described in (**A**) were made, analyzed by SDS-PAGE and immunoblotting, and detected using an anti-GFP antibody. The REVERT total protein stain of the membrane is shown as a loading control. Molecular weights are shown on the left side in kDa.

**Figure 4 biomolecules-12-00533-f004:**
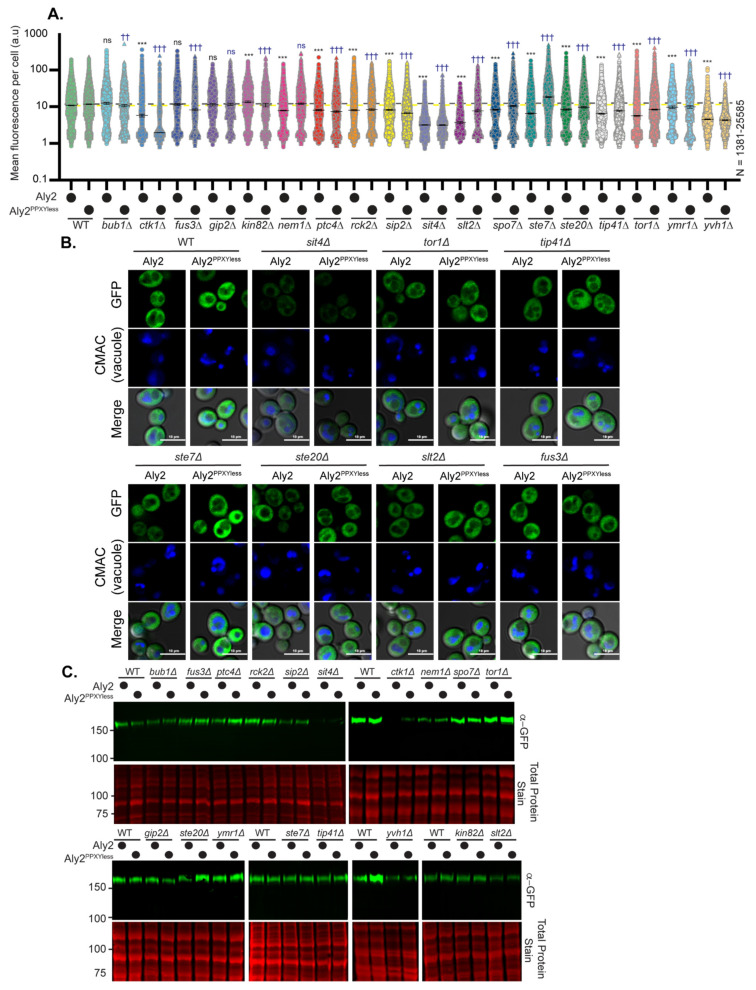
Kinases and phosphatases from the KinDel screen alter the abundance and electrophoretic mobility of Aly2: (**A**) Cells expressing the indicated Aly2 protein fused to GFP (pRS415-*TEF1pr* plasmids) in either WT or the noted gene deletion backgrounds were imaged by high-content confocal microscopy, and whole-cell fluorescence of the cells was quantified using NIS.*ai* and Nikon GA3 software. The mean fluorescence intensity from whole-cell measurements (in arbitrary units, au) is plotted for each cell as a circle. The median fluorescence intensity is shown as a black line for each group and the error bars represent the 95% confidence interval. A yellow or black dashed line represents the median fluorescence intensity for Aly2 or Aly2^PPXYless^ expressed in WT cells, respectively. Kruskal–Wallis statistical analysis with Dunn’s post hoc test was performed to compare the fluorescence distributions to the cognate WT. In black asterisks (*) or blue daggers (†), comparisons are made to Aly2 or Aly2^PPXYless^ in WT cells, respectively (ns = not significant; three symbols have a *p*-value < 0.0005). (**B**) A subset of the fluorescent microscopy images acquired for the data presented in (**A**) are shown. CMAC is used to stain the vacuoles (shown in blue). Merge is overlaid with the transmitted light cell image as well as the fluorescence images. (**C**) Whole-cell extracts from the cells described in (**A**) were made, analyzed by SDS-PAGE and immunoblotting, and detected using an anti-GFP antibody. The REVERT total protein stain of the membrane is shown as a loading control. Molecular weights are shown on the left side in kDa.

**Figure 5 biomolecules-12-00533-f005:**
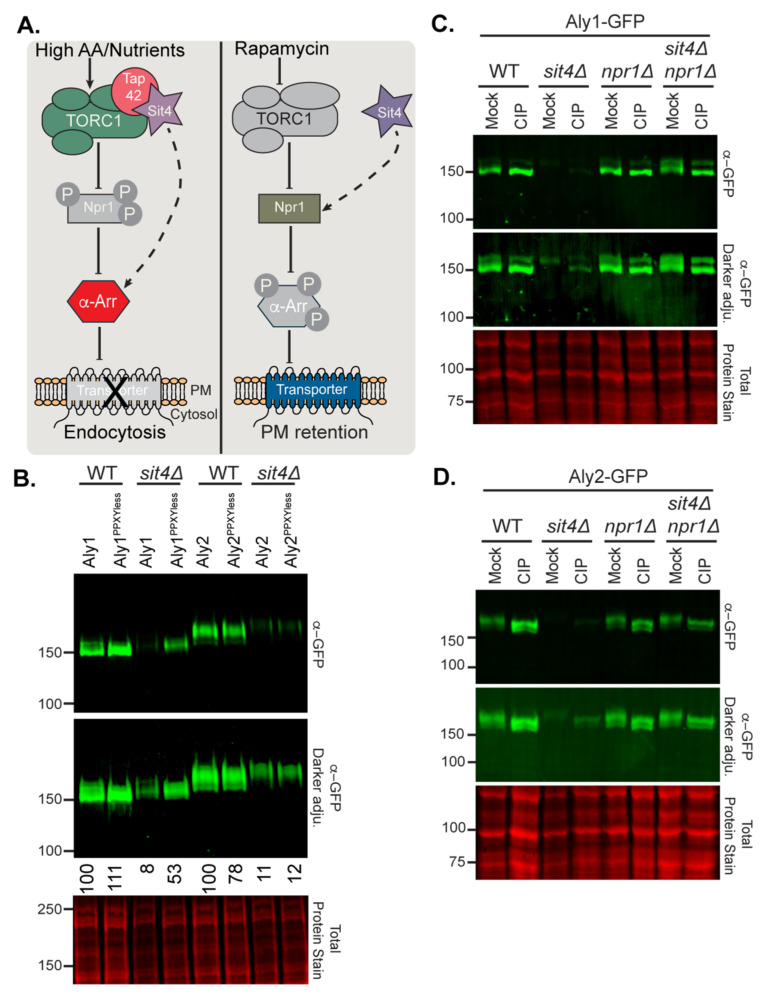
Sit4 and Npr1 regulate Aly abundance and phosphorylation: (**A**) Model of TORC1 regulation of Sit4 and Npr1 based on studies of Bul regulation of Gap1. In robust nutrients, TORC1 is active and Tap42-Sit4 are thought to be bound. This may direct Sit4 activity towards α-arrestins [12,13], as indicated by a dashed line. α-Arrestins are active and can remove nutrient transporters from the PM. In response to rapamycin or poor nutrient conditions, TORC1 is inhibited, and the Tap42-Sit4 complex dissociates. Sit4 phosphatase activity is now thought to be directed to Npr1 (dashed-line with arrow), which becomes active. Active Npr1 can phosphorylate and inhibit α-arrestins, preventing the removal of nutrient transporters from the PM. Activation or inhibition are indicated by lines that end in arrowheads or blunt lines, respectively. Complexes whose activity is lost in the condition indicated are shown in grey. (**B**) Whole-cell extracts were made from the indicated strains expressing Alys as GFP-tagged proteins from the *TEF1pr* and then resolved by SDS-PAGE. Immunoblots were probed with anti-GFP antibody to examine Aly abundance and mobility. REVERT total protein stain is shown for each blot as a loading control (red). The percent of Aly protein, corrected for the loading control, for each lane is calculated relative to WT Aly in WT cells. (**C**,**D**) Whole-cell extracts were made from the indicated strains expressing Aly-GFP from the *TEF1pr* and then were either treated with calf intestinal alkaline phosphatase (CIP) or incubated in the same conditions without enzyme (mock). Extracts were then resolved by SDS-PAGE and probed with anti-GFP antibody. REVERT total protein stain is shown for each blot as a loading control (red). For all panels, molecular weights are shown on the left side in kDa.

**Figure 6 biomolecules-12-00533-f006:**
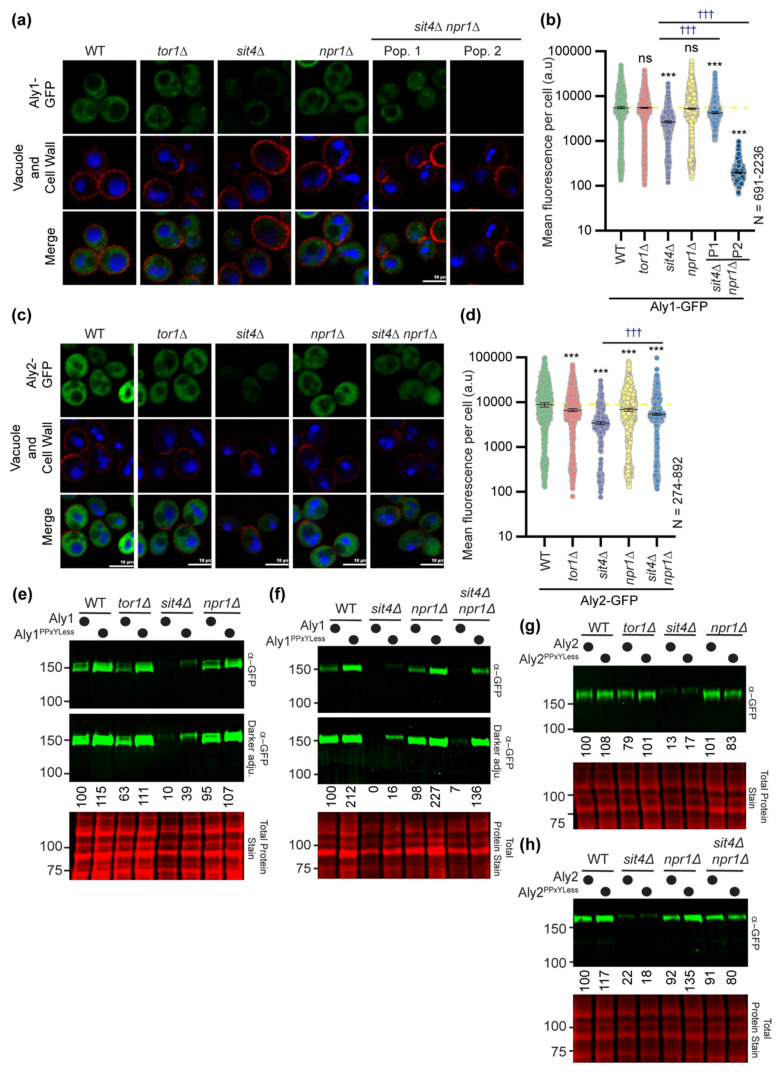
Sit4 and Npr1 regulate Aly abundance but do not affect localization: (**a**) Aly1-GFP or (**c**) Aly2-GFP was expressed from the *TEF1pr* in either WT cells or those lacking the gene indicated and imaged by high-content fluorescence microscopy. CMAC is used to stain the vacuoles (shown in blue) and trypan blue (shown in red) is used to mark the cell wall. All images are shown as equally adjusted from a single experiment. (**b**) or (**d**) The whole-cell fluorescence of cells imaged in (**a**) or (**c**), respectively, was determined using NIS.*ai* and Nikon GA3 software, and the fluorescence for each cell is plotted as a circle. The median fluorescence intensity is shown as a black line for each group and the error bars represent the 95% confidence interval. A yellow dashed line represents the median fluorescence intensity for Aly1 or Aly2, expressed in WT cells in panels b and d, respectively. Kruskal–Wallis statistical analysis with Dunn’s post hoc test was performed to compare the fluorescence distributions to the cognate WT. In black asterisks (*), statistical comparisons are made to Aly1 or Aly2 in WT cells for panels (**b**) and (**d**), respectively. In blue daggers (†), the indicated statistical comparisons are made (ns = not significant; three symbols = *p*-value < 0.0005). (**e**–**h**) Whole-cell extracts were made from the indicated strains expressing Alys as GFP-tagged proteins from the *TEF1pr* and then resolved by SDS-PAGE. Immunoblots were probed with anti-GFP antibody to examine Aly abundance and mobility. REVERT total protein stain is shown for each blot as a loading control (red). The percent of Aly protein, corrected for the loading control, for each lane is calculated relative to WT Aly in WT cells.

**Figure 7 biomolecules-12-00533-f007:**
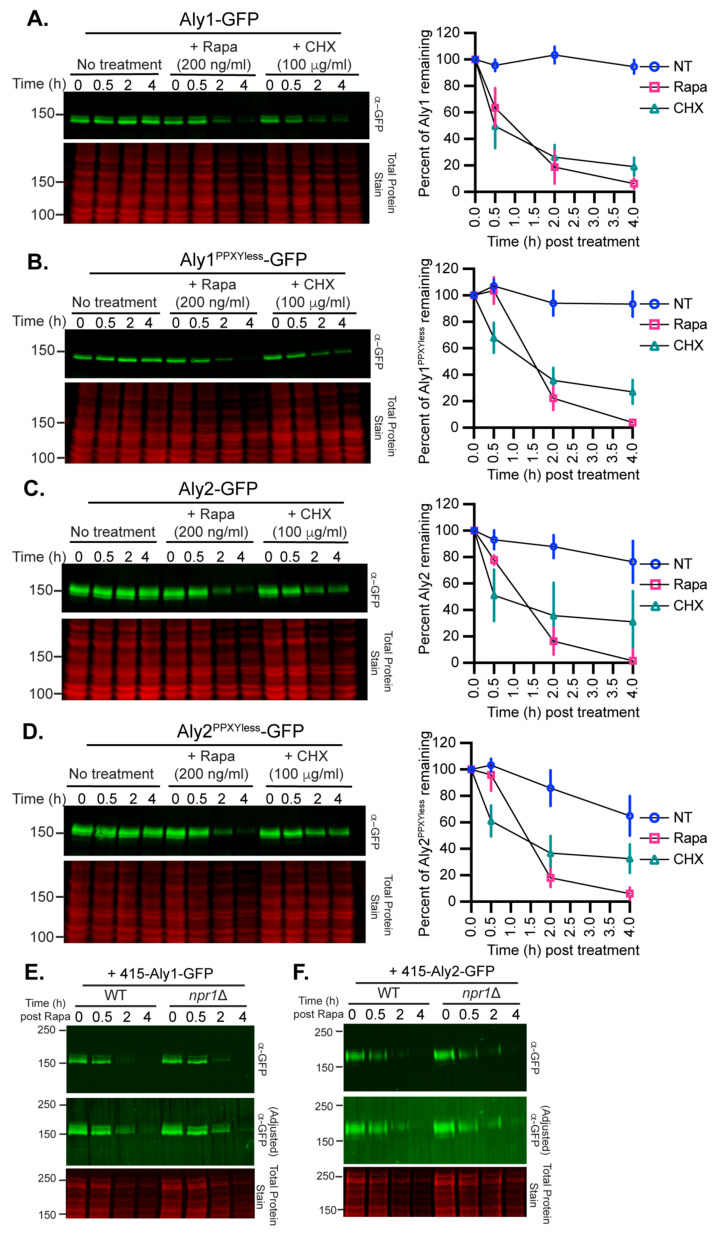
Rapamycin treatment destabilizes Aly1 and Aly2, irrespective of Rsp5 binding: (**A**–**D**) Whole-cell extracts from WT cells treated with nothing (no treatment; NT), rapamycin (rapa 200 ng/mL), or cycloheximide (CHX 100 μg/mL) for the times indicated, and expressing (**A**) Aly1-GFP, (**B**) Aly1^PPXYless^-GFP, (**C**) Aly2-GFP, or (**D**) Aly2^PPXYless^-GFP, or from *npr1*∆ cells expressing (**E**) Aly1-GFP or (**F**) Aly2-GFP, were resolved by SDS-PAGE. Anti-GFP antibody was used to detect tagged α-arrestins and REVERT total protein stain was used as a loading control. Blots shown are one representative from 4 replicate experiments. Molecular weights are shown on the left side in kDa. For blots in panels (**A**–**D**), the pixel intensities for the GFP-detected band and the lane in the loading control were measured using Image J. A correction factor based on the loading control was applied to each pixel intensity measurement. The t = 0 point was set to 100%, and all abundances are presented relative to that point. The error bars represent the SEM.

**Figure 8 biomolecules-12-00533-f008:**
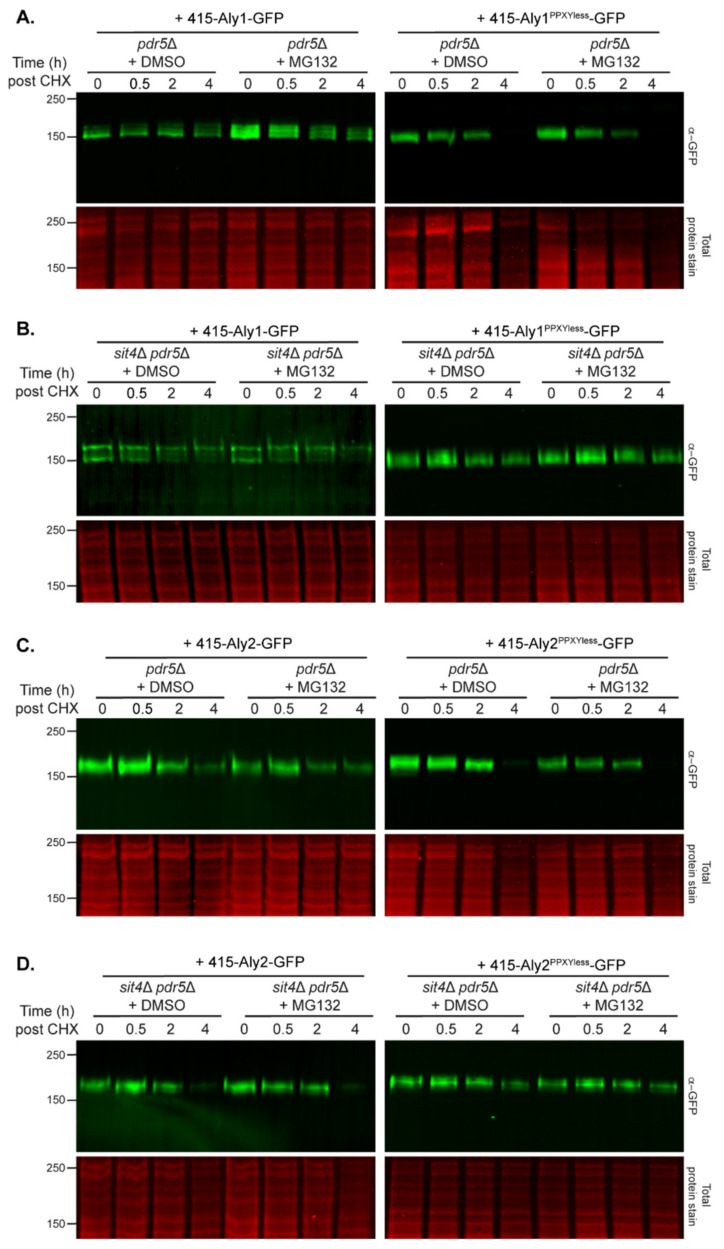
Inhibition of the proteasome does not lead to stabilization of Alys: (**A**–**D**) Whole-cell extracts from cells expressing (**A**) Aly1-GFP, (**B**) Aly1^PPXYless^-GFP, (**C**) Aly2-GFP, or (**D**) Aly2^PPXYless^-GFP were treated with either vehicle control (DMSO) or MG132 for 60 min and then cycloheximide (CHX) for the time indicated (hours); proteins were resolved by SDS-PAGE. Anti-GFP antibody was used to detect tagged α-arrestins and REVERT total protein stain was used as a loading control. Blots shown are one representative from 2–3 replicate experiments. Molecular weights are shown on the left side in kDa.

**Figure 9 biomolecules-12-00533-f009:**
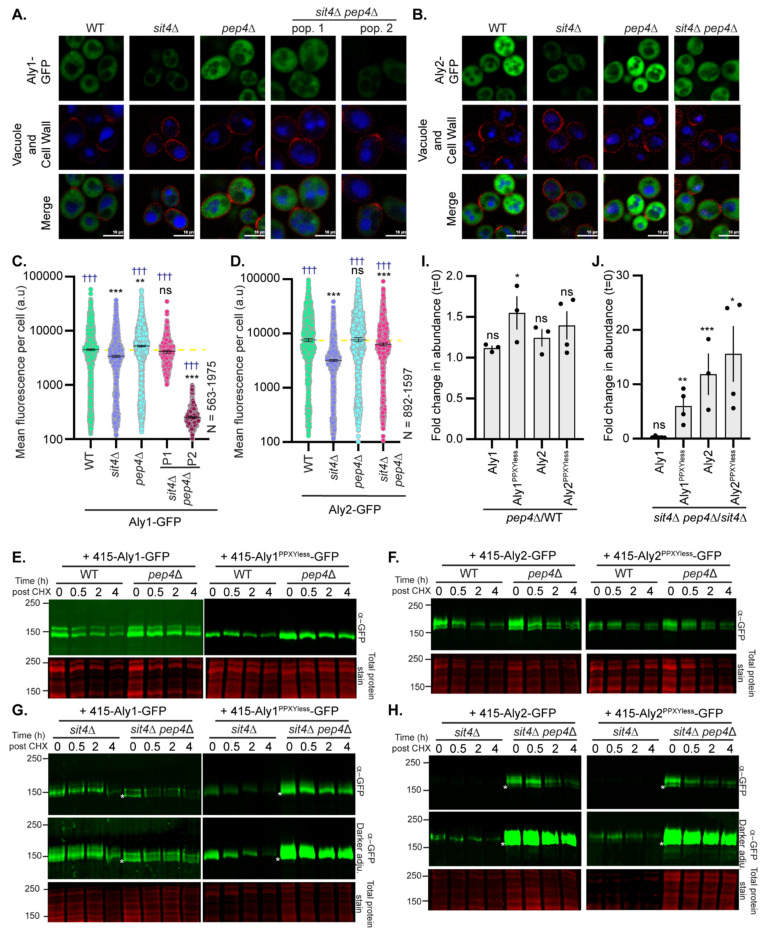
Impaired vacuole protease function increases Aly protein abundance: (**A**) Aly1-GFP or (**B**) Aly2-GFP was expressed from the *TEF1pr* in either WT cells or those lacking the gene indicated and imaged by high-content fluorescence microscopy. CMAC is used to stain the vacuoles (shown in blue), and trypan blue (shown in red) is used to mark the cell wall. (**C**) or (**D**) The whole-cell fluorescence of cells imaged in (**A**) or (**B**), respectively, was determined using NIS.*ai* software, and the fluorescence for each cell is plotted as a circle. The median fluorescence intensity is shown as a black line for each group, and the error bars represent the 95% confidence interval. A yellow dashed line represents the median fluorescence intensity for Aly1 or Aly2, expressed in WT cells in panels (**G**) and (**H**), respectively. Kruskal–Wallis statistical analysis with Dunn’s post hoc test was performed to compare the fluorescence distributions to the cognate WT. In black asterisks (*), statistical comparisons are made to Aly1 or Aly2 in WT cells for panels (**C**,**D**), respectively. In blue daggers (†), statistical comparisons are made to Aly1 or Aly2 in *sit4*∆ cells for panels (**C**,**D**), respectively (ns = not significant; two symbols = *p*-value < 0.005; three symbols = *p*-value < 0.0005). (**E**–**H**) Whole-cell extracts expressing the α-arrestin indicated in WT, *pep4*∆, *sit4*∆, or *sit4*∆ *pep4*∆ cells were made from cells treated with cycloheximide (CHX) for the time indicated (hours) and resolved by SDS-PAGE. Anti-GFP antibody was used to detect tagged α-arrestins and REVERT total protein stain was used as a loading control. Blots shown are one representative from 2–3 replicate experiments. Molecular weights are shown on the left side in kDa. For panels (**G**,**H**), a white asterisk marks the faster migrating species observed in the *sit4*∆ *pep4*∆ extracts that are not found in the *sit4*∆ alone. (**I**,**J**) A plot of the fold change in the t = 0 absolute pixel intensities for the Aly-GFP bands shown in (**E**,**F**) or (**G**,**H**), respectively, is presented so that the influence of Pep4 on steady-state Aly abundances can be considered relative to WT (**E**,**F**,**I**) or *sit4*∆ (**G**,**H**,**J**) cells. Each fold change data point is presented as a circle, and the mean is represented by the height of the bar graph. Error bars represent the SEM. Student’s *t*-test with Welch’s correction was performed on the pixel intensity values for each (i.e., comparing Aly1 t = 0 band intensities for WT vs. *pep4*∆), and results are presented over each column. ns = not significant; * = *p* < 0.05; ** = *p* < 0.005; *** = *p*-value < 0.0005).

**Figure 10 biomolecules-12-00533-f010:**
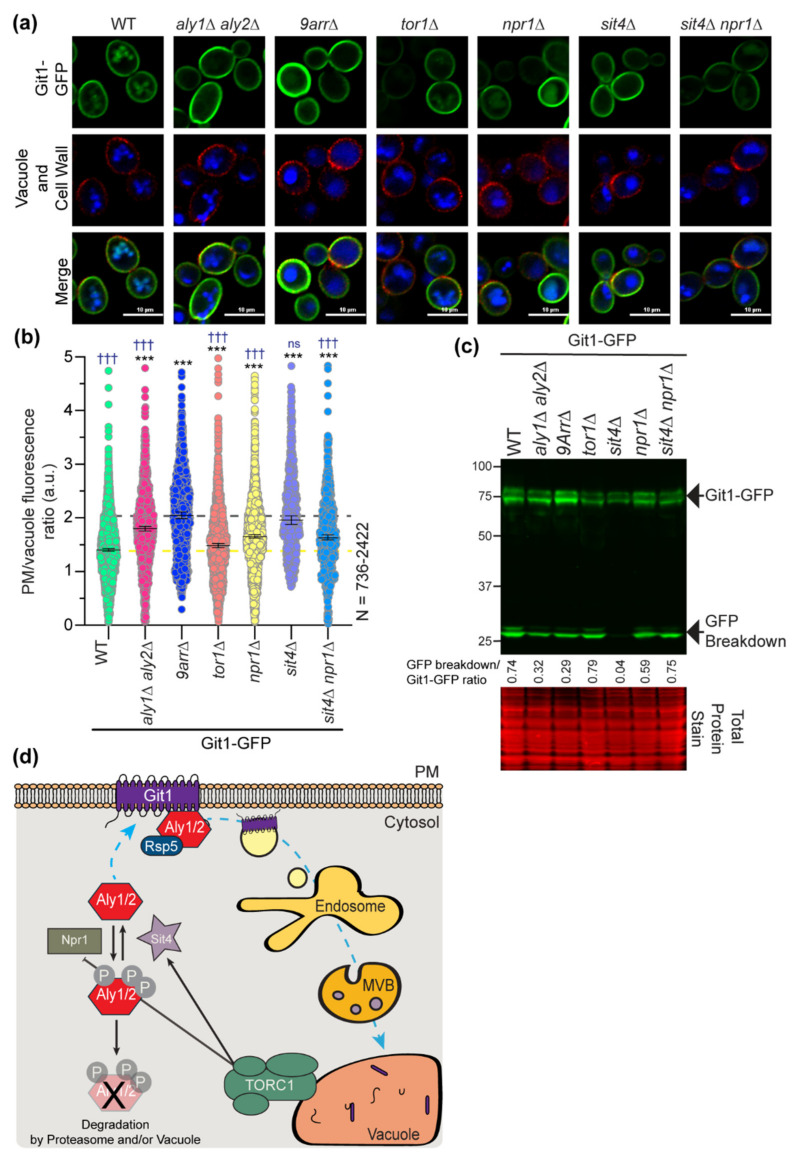
TORC1-Sit4-Npr1 regulation of Aly-mediated trafficking of Git1: (**a**) Git1-GFP was expressed from the *TEF1pr* in either WT cells or those lacking the gene indicated and imaged by high-content fluorescence microscopy. CMAC is used to stain the vacuoles (shown in blue) and trypan blue (shown in red) is used to mark the cell wall. (**b**) The PM and vacuolar fluorescence intensities from the cells depicted in (**a**) were quantified using NIS.*ai* and Nikon GA3 software and the distributions of the PM/vacuole fluorescence ratios in arbitrary units (a.u.) were plotted as scatter plots. The horizonal midline in black represents the median and the 95% confidence interval is represented by the error bars. Yellow and grey dashed lines are used as references to indicate the median ratio for WT or 9Arr∆ cells, respectively. Kruskal–Wallis statistical analysis with Dunn’s post hoc test was performed to compare the fluorescence distributions to either WT or 9Arr∆ cells. In black asterisks (*), statistical comparisons are made to WT cells. In blue daggers (†), statistical comparisons are made to 9Arr∆ cells. (ns = not significant; three symbols = *p*-value < 0.0005). (**c**) Whole-cell extracts from cells expressing Git1-GFP from the *TEF1* promoter were resolved by SDS-PAGE and immunoblotted. Anti-GFP antibody was used to detect Git1-GFP and REVERT total protein stain was used as a loading control. Blot shown is representative of 2 replicate experiments. Molecular weights are shown on the left side in kDa. The ratio of GFP breakdown product (represents the vacuolar pool of GFP) over Git1-GFP (represents the pool of Git1 outside of the vacuole) for each lane is presented. (**d**) Model of TORC1-Sit4-Npr1 regulation of Aly-mediated trafficking of Git1.

**Table 1 biomolecules-12-00533-t001:** Summary of screening 18 gene deletion candidates from KinDel library with over-expression of Aly1 or Aly2.

	pRS415-*TEF1pr*-Aly1-GFP				pRS415-*TEF1pr*-Aly2-GFP			
Gene	NaCl	Rapa	Protein Levels	Mobility Shift?	NaCl	Rapa	Protein Levels	Mobility Shift?
*bub1*∆	Sens	Sens	Equal	No	Sens	Sens	Equal	No
*ctk1*∆	Sens	Sens	Lower	No	Sens	Sens	Lower	Maybe
*fus3*∆	Sens	Res	Hi/Equal	No	Sens	Equal	Equal	No
*gip2*∆	Sens	Res	Equal	No	Sens	Res	Equal	Yes
*kin82*∆	Equal	Equal	Equal	No	Equal	Res	Equal	No
*nem1*∆	Equal	Sens	Lower	No	Equal	Sens	Lower	No
*ptc4*∆	Equal	Equal	Equal/Lower	No	Equal	Equal	Equal/Lower	No
*rck2*∆	Equal	Equal	Equal/Lower	No	Equal	Equal	Equal/Lower	No
*sip2*∆	Res	Sens	Lower	Yes	Res	Sens	Lower	No
*sit4*∆	Sens	Sens	Lower	Yes	Sens	Res	Lower	Yes
*slt2*∆	Sens	Sens	Equal	No	Equal	Sens	Lower	No
*spo7*∆	Equal	Sens	Lower	No	Res	Sens	Equal/Lower	No
*ste7*∆	Res	Res	Equal/Lower	No	Res	Res	Equal/Lower	No
*ste20*∆	Sens	Sens	Lower	No	Sens	Sens	Equal/Lower	Yes
*tip41*∆	Equal	Res	Lower	No	Equal	Res	Equal/Lower	No
*tor1*∆	Equal	Sens	Equal/Lower	No	Equal	Sens	Equal/Lower	No
*ymr1*∆	Equal	Res	Equal/Lower	No	Equal	Equal	Equal	No
*yvh1*∆	Sens	Sens	Lower	Yes	Sens	Sens	Lower	Yes

## Data Availability

The data presented in this study are available in this article or in the Appendix A.

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
