# Peer review of "TORC1 Signaling Controls the Stability and Function of α-Arrestins Aly1 and Aly2"

_biomolecules, 2022, doi:10.3390/biom12040533_

Round 1
Reviewer 1 Report
The manuscript of Bowman and colleagues details a comprehensive analysis of how the Rsp5 adaptor proteins Aly1 and Aly2 in budding yeast are regulated by post-translational modulation of their phosphorylation status. A screen was performed in cells over-expressing each of the Aly a-arrestins, evaluating the contribution of almost 200 phopho-enzymes to regulation, and incorporating physiological stress conditions of rapamycin treatment and high salt.
A subset of interesting mutants are then followed up, with a particular focus on how the TORC1 pathway, and the related factors Npr1 and Sit4, regulates and the surface protein landscape via Aly-proteins.
The authors should be commended for the level of thoroughness achieved in this paper, not only are phenotypes quantified across large numbers, but complementary analyses are included to back up the results, most of which are very convincing.
I do not think many changes are required for this article to be published, and I believe this will be an important paper and a valuable resource for future studies. I have made some comments for consideration, detailed below.
Major points
- The paper is well written, and the use of language makes for a particularly enjoyable read. The references are thorough and appropriate. However, the manuscript seems unnecessarily long. This could be reduced significantly by:
- Removing redundant text, which occurs in places across the introduction, methods, results, and discussion. For example, the ‘known phosphorylation sites of Alys’ and ‘details of KinDel library’ are described in detail, almost verbatim, in different sections.
- Moving the yeast/plasmids reagent information to a supplemental table, since the journal typeset forces this across ~5 pages.
- On page 15, line 464 the authors state that Alys trafficking functions are important for rapamycin and salt stress, as the phenotypes observed rely on PPxY motifs. It seems important to comment here (or in the discussion) that components of the TORC1 signalling pathway are amongst the weakest to show PPxY dependence in Figure 2. Compared with black/white examples like fus3∆ rapamycin and sip2∆ salt, the TORC1 representatives (tor1∆, tip41∆, and sit4∆) have more modest differences, or sometimes Aly-specific differences. It is especially important to acknowledge/discuss as the latter parts of the manuscript deal specifically with TORC1 related functions. Maybe discussing rapamycin and salt growth phenotypes separately would better highlight the important differences?
- From page 30, line 862 - discussing the differences of WT, npr1∆ and tor1∆, from only microscopy (Fig 10A and 10B) and then later immunoblot (Fig 10C) might be better discussed by combining the observations. The microscopy differences of PM Git1-GFP between WT and tor1∆ might be statistically significant but when considered with the blots might not be biologically significant (if so, this could be tapered in the discussion). An idea in relation to this might be to assess Git1-GFP trafficking in another TORC1 mutant (tco89∆), or in pdr5∆ treated with rapamycin and then included.
Minor points
- There is a missing reference (Robinson et al 2021) on page 3, line 139.
- I am not sure the sentence “Gene deletions that altered Aly1 or Aly2 growth significantly” on Page 9, line 226 makes sense? Do you maybe mean “altered levels of Aly1/2” or “altered growth of cells over-expressing Aly1/2)?
- The serial dilutions of yeast cultures used in growth assays is described as either 3- or 5- fold in the methods section but what fold-dilution used for particular experiments (e.g. page 13, line 433) is not mentioned in the legends.
- Page 20, line 599 - the distribution (just as a %) between the two populations should be documented.
Reviewer 2 Report
The manuscript describes an interesting regulatory system that links TORC1 signaling via the alpha-arrestins Aly1/2 to the regulation of Git1 stability. The data show convincingly how two outputs of TORC1, Npr1 and Sit4, work in parallel to accomplish this regulation. The experiments are of high quality and the data are well presented and discussed. One issue that should be addressed is that the model is presented as universal models of arrestin-mediated regulation of nutrient transporters. For example, the model in Figure 5a or the text in the Discussion (“There is a robust literature connecting alpha-arrestins to TORC1 signaling and these studies largely converge on a model wherein cells with active TORC1, alpha-arrestins are dephosphorylated and can mediate endocytosis of nutrient permeases.”) generalizes this regulation, which is not correct. The presented model fits the Gap1 studies but does not fit with many other studies that have shown rapid degradation of nutrient transporters during TORC1 shutdown (amino acid starvation or rapamycin treatment). In this regard, Gap1 seems to be the exception, being regulated in the reverse way compared to high affinity amino acid transporters (Can1, Tat2, Bap2, Mup1). For example, Shmidt et al. (EMBO J 1998) demonstrated that Tat2 is downregulated during TORC1 shutdown and that Tat2 is stabilized during high TORC1 signaling in an Npr1 dependent way. The MacGurn study published that Can1 is stabilized when TORC1 is inactive (rapamycin treatment), which is not correct and one of the reasons why there is confusion in the field (e.g., see Jones CB et al., Traffic 2012). Therefore, I recommend that the authors rephrase the sections discussing the TORC1-Npr1-Sit4-arrestin model and acknowledge that the model only explains the regulation of a set of transporters but cannot be generalized.
